
# Statistical Analysis for Satellite Index-Based Insurance to define Damaged Pasture Thresholds

Juan José Martín-Sotoca[1]*, Antonio Saa-Requejo[2,3], Rubén Moratiel[2,3], Nicolas Dalezios[4], Ioannis Faraslis[5], and Ana María Tarquis[2,6]

jmartinsotoca@gmail.com,       antonio.saa@upm.es,       ruben.moratiel@upm.es,       dalezios.n.r@gmail.com,
faraslisgiannis@yahoo.gr, anamaria.tarquis@upm.es

[1] Data Science Laboratory. European University, Madrid, Spain.
[2] CEIGRAM, Research Centre for the Management of Agricultural and Environmental Risks, Madrid, Spain.
[3] Dpto. Producción Agraria. Universidad Politécnica de Madrid, Spain.
[4] Department of Civil Engineering. University of Thessaly, Volos, Greece.
[5] Department of Planning and Regional Development. University of Thessaly, Volos, Greece.
[6] Grupo de Sistemas Complejos. Universidad Politécnica de Madrid, Spain.

*  Correspondence to: jmartinsotoca@gmail.com

**Abstract:** Vegetation indices based on satellite images, such as Normalized Difference Vegetation Index (NDVI), have been used in countries like USA, Canada and Spain for damaged pasture and forage insurance for the last years. This type of agricultural insurance is called "satellite index-based insurance" (SIBI). In SIBI, the occurrence of damage is defined through NDVI thresholds mainly based on statistics derived from normal distributions. In this work a pasture area at the north of Community of Madrid (Spain) has been delimited by means of MODIS images. A statistical analysis of NDVI histograms was applied to seek for the best statistical distribution using maximum likelihood method. The results show that the normal distribution (NORMAL) is not the optimal representation and the General Extreme Value (GEV) distribution presents a better fit through the year. A comparison between NORMAL and GEV are showed respect to the probability under a NDVI threshold value along the year. This suggests that a priori distribution should not be selected and a percentile methodology should be used to define a NDVI damage threshold rather than the average and standard deviation, typically of normal distributions.

**Keywords:** NDVI, pasture insurance, GEV distribution, MODIS.

## Highlights

- **General Extreme Value (GEV) distribution provides the best fit to the NDVI historical observations.**

- **Difference between Normal and GEV distributions are higher during spring and autumn, transition periods in the precipitation regimen.**

- **NDVI damage threshold shows evident differences using Normal and GEV distributions covering both the same probability (24.20%).**

- **NDVI damage threshold values based on percentiles calculation is proposed as an improvement in the index based insurance in damaged pasture.**






## 1. Introduction

Agricultural insurance addresses the reduction of the risk associated with crop
production and animal husbandry. The concept of index-based insurance (IBI) attempts
to achieve settlements based on the value taken by an objective index rather than on a
case-by-case assessment of crop or livestock losses (Gommes and Kayitakier, 2013).
Indeed, the goal of IBI policy remains to develop an affordable tool to all producers,
including smallholders. Specifically, IBI can constitute a safety net against
weather-related risks for all members of the farming community, thereby increasing
food security and reducing the vulnerability of rural populations to weather extremes.
Moreover, IBI can be associated with credits for insured smallholders, due to the fact
that the risk of non-repayment for lenders is reduced, which encourages the use of
agricultural inputs and equipment, leading to increased and more stable crop
production. Over the past decade, the importance of weather index-based insurances
(WIBI) for agriculture has been increasing, mainly in developing countries (Gommes
and Kayitakier, 2013). This interest can be explained by the potential that IBI
constitutes a risk management instrument for small farmers. Indeed, it can be
considered within the context of renewed attention to agricultural development as
one of the milestones of poverty reduction and increased food security, as well as the
accompanying efforts from various stakeholders to develop agricultural risk
management instruments, including agricultural insurance products.

Farmers need to protect their land and crops specifically from drought in arid and
semi-arid countries, since their production may directly depend mainly on the impacts
of this particular natural hazard. Insurance for drought-damaged lands and crops is
currently the main instrument and tool that farmers can resort in order to deal with
agricultural production losses due to drought. Many of these insurances are using
satellite vegetation indices (Rao, 2010), thus they are also called "satellite index-based
insurances" (SIBI). SIBI have some advantages over WIBI, such as cost-effective
information and acceptable spatial and temporal resolution. They do not, however,
resolve the issue of basis risk, i.e. potential unfairness to insurance takers (Leblois,
2012). Moreover, the very nature of an index-based product creates the chance that
an insured party may not be paid when they suffer loss. For this reason, in some
countries (Spain) they have named this SIBI as "damaged in pasture" to cover not only
drought even this one is the main cause.

It is highly recognized that shortage of water has many implications to agriculture,
society, economy and ecosystems. Specifically, its impact on water supply, crop
production and rearing of livestock is substantial in agriculture. Knowing the likelihood
of drought is essential for impact prevention (Dalezios, 2013). Drought severity



assessment can be approached in different ways: through conventional indices based
on meteorological data, such as temperature, rainfall, moisture, etc. (Niemeyer, 2008),
as well as through remote sensing indices based on images usually taken by artificial
satellites (Lovejoy et al., 2008) or drones. In the second group they are found Satellite
Vegetation Indices (SVI), which can quantify "green vegetation", and soil moisture
through Soil Water Index (Gouveia et al., 2009) combining different spectral
reflectances. Thus, they are one of the main ways to quantitatively assess drought
severity.
At the present time, several satellites (NOAA, TERRA, DEIMOS, etc.) can provide
this spectral information with different spatial resolution. Some series with a high
temporal frequency are freely available, those from NOAA satellites and Terra. The
most widely known SVI is the Normalized Difference Vegetation Index (NDVI). It
follows the principle that healthy vegetation mainly reflects the near-infrared
frequency band. There are several other important SVI, such as Soil Adjusted
Vegetation Index (SAVI) and Enhanced Vegetation Index (EVI) that incorporate soil
effects and atmospheric impacts, respectively. An important point of this class of
insurance is "when damage occurs". To measure this, a SVI threshold value is defined
mainly based on statistics that apply to normally distributed variables: average and
standard deviation. When current SVI values are bellow this threshold value for a
period of time, insurance recognizes that a damage is occurring, most of the times
drought, and then it begins to pay compensations to farmers.
WIBI aims to protect farmers against weather-based disasters such as droughts,
frosts and floods. A WIBI policy links possible insurance payouts with the weather
requirements of the crop being insured: the insurer pays an indemnity whenever the
realized value of the weather index meets a specified threshold. Whereas payouts in
traditional insurance programs are related to actual crop damages, a farmer insured
under a WIBI contract may receive a payout. A current difficulty to the wide
implementation of WIBI is the weakness of indices. Indeed, there is certainly a need for
more efficient indices based on the additional experience gained from the
implementation of WIBI products in the developing world. Current trends in index
technology are exciting and they actuate high expectations, especially the
development of yield indices and the use of remote sensing inputs. Risk protection and
insurance illiteracy constitute another difficulty, which has to be addressed by training
and awareness-raising at all levels, from farmers to farmers' associations,
micro-insurance partners, as well as senior decision-makers in insurance, banking, and
politics (Bailey, 2013). It is essential that all stakeholders (especially the insured)
perfectly understand the principles of IBI, as otherwise the insurer, even the whole
concept of insurance, is at risk of reputation loss for years or decades.



There is currently a lack of technical capacity in the insurance sectors of most
developing countries, which is a constraint to the scaling up and further development
of WIBI (Gommes and Kayitakire, 2012). Specifically, although it is possible to design an
index product and assist in roll-out, marketing, and sales, such assistance is not
possible on a wide scale, simply because there is lack of qualified expertise. Indeed, it
usually requires mathematical modeling, data manipulation, and expertise in crop
simulation to design an index. Nevertheless, it is possible to structure insurance with
multiple indices, but this increases the complexity of the product and makes it difficult
for farmers to comprehend it. 'Basis risk' is also a particular problem for index
products, which is frequently caused by the fact that measurements of a particular
variable, such as rain, may differ at the insurer's measurement site and in the farmer's
field. This also creates problems for insurance providers. Indeed, part of the reason the
scaling up of index products has failed is that both insurers and farmers suffer from
this basis risk.

Currently, to mitigate impacts of climate-related reduced productivity of French
grasslands, several studies have been developed to design new insurance scheme
bases indemnity payouts to farmers on a forage production index (FPI) (Rumiguié et
al., 2015; 2017). Two examples of SIBIs are presented in two different countries: USA
and Spain. In particular, in USA there are several insurance programs for pasture,
rangeland and forage, which use various indexing systems (rainfall and vegetation
indices), and are promoted by Unites States Department of Agriculture (USDA) (Maples
et al., 2016; USDA, 2018). NDVI is the index chosen in the vegetation index program
and it is obtained from AVHRR (Advanced Very High Resolution Radiometer) sensor
onboard NOAA satellites. Average, maximum and minimum NDVI values are obtained
from a historical series with the aim of calculating a trigger value. Insurer decides the
quantity of compensation comparing this trigger with current value. On the other
hand, in Spain there exists the "Insurance for Damaged Pasture" from "Spanish System
of Agricultural Insurance" (BOE, 2013). This insurance defines damage event through
NDVI values obtained from MODIS (Moderate Resolution Imaging Spectroradiometer)
sensor onboard TERRA satellite of NASA. In this insurance, NDVI threshold values
($NDVI_{th}$) are calculated subtracting several times ($k = 0.7\ or\ k = 1.5$) standard deviation
to average within a homogeneous area:

$$NDVI_{th} = \mu - k \cdot \sigma \tag{1}$$


where $\mu, \sigma$ are average and standard deviation of NDVI respectively. Average and
standard deviation come of supposing normal distributions in the historical data
(Goward et al., 1985; Hobbs, 1995; Fuller, 1998; Al-Bakri and Taylor, 2003; Turvey et
al., 2012; De Leeuw et al. 2014).



The aim of this paper is to find the best statistical NDVI distribution without the "a
priori" assumption that variables follow a Normal distribution, typically for current SIBI
methodology. In order to achieve this, the Maximum Likelihood Method (MLM) is
fitted to a historical series of NDVI values in a pasture land area in Spain (Community
of Madrid). Different types of distributions are examined with the aim of finding the
best fit. To eliminate some noise in the historical series, an original method is applied
consisting of using Hue-Saturation-Lightness (HSL) color model. Finally, Chi-square test
($\chi^2$ test) has been used to check the goodness of fit for all considered distributions.

## 2. Materials and Methods
### *2.1 Vegetation Index*
The differences of the reflectance of green vegetation in parts of the
electromagnetic radiation spectrum, namely, visible and near infrared, provide an
innovative method for monitoring surface vegetation from space. Specifically, the
spectral behavior of vegetation cover in the visible (0.4-0.7mm) and near infrared
(0.74-1.1mm, 1.3-2.5mm) offers the possibility to monitor from space the changes in
the different stages of cultivated and uncultivated plants taking also into account the
corresponding behavior of the surrounding microenvironment (Ortega-Farias et al.,
2016). Indeed, from the visible part of the electromagnetic radiation spectrum it is
possible to draw conclusions about the rate photosynthesis, whereas from near
infrared inferences are extracted about the chlorophyll density and the amount of
canopy in the plant mass, as well as the water content in the leaves, which is also
linked directly to the rate of transpiration with impacts to physiological process of
photosynthesis. Usually, data from NOAA/AVHRR series of polar orbit meteorological
satellites are used with low spatial resolution (1.1 km$^2$) and recurrence interval at least
twice daily from the same location. Several algorithms combining channels of red
(RED), near infrared (NIR) and green (GREEN) have been proposed, which provide
indices sensitive to green vegetation.
NDVI uses two frequency bands: red band (660 nm) and near-infrared band (860
nm). Absorption of red band is related to photosynthetic activity and reflectance of
near-infrared band is related to presence of vegetation canopies (Flynn, 2006). In
drought periods, NDVI values can reduce significantly, therefore many researchers
have used this index to measure drought events in recent years (Dalezios et al., 2014).
To calculate NDVI we will use this mathematical formula:

$$NDVI = \frac{IR-R}{IR+R} \tag{2}$$




where IR and R are reflectance values in Near-Infrared band and Red band,
respectively. NDVI values below zero indicate no photosynthetic activity and are
characteristic of areas with large accumulation of water, such as rivers, lakes, or
reservoirs. The higher is the NDVI value, the greater is the photosynthetic activity and
vegetation canopies.

In this paper, the NDVI is used, which is widely known index with a multitude of
applications over time. The NDVI is suited for monitoring of total vegetation, since it
partly compensates the changes in light conditions, land slope and field of view (Kundu
et al., 2016). In addition, clouds, water and snow show higher reflectance in the visible
than in the near infrared, thus, they have negative NDVI values. Indeed, bare and rocky
terrain show vegetation index values close to zero. Moreover, the NDVI constitutes a
measure of the degree of absorption by chlorophyll in the red band of the
electromagnetic spectrum. In summary, the NDVI is a reliable index of the chlorophyll
density on the leaves, as well as the percentage of the leaf area density over land,
thus, NDVI constitutes a credible measure for the assessment of dry matter (biomass)
in various species vegetation cover (Dalezios, 2013). It is clear from the above that the
NDVI is an index closely related to growth and development of plants, which can
effectively monitor surface vegetation from space.

The continuous increase of the NDVI value during the growing season reflects the
vegetative and reproductive growth due to intense photosynthetic activity, as well as
the satisfactory correlation with the final biomass production at the end of a growing
period. On the other hand, gradual decrease of the NDVI values signifies stress due to
lack of water or extremely high temperatures for the plants, leading to a reduction of
the photosynthetic rate and ultimately a qualitative and quantitative degradation of
plants. NDVI values above zero indicate the existence of green vegetation
(chlorophyll), or bare soil (values around zero), whereas values below zero indicate the
existence of water, snow, ice and clouds.

**2.2 Database**
Scientific research satellite Terra (EOS AM-1) has been chosen to provide
necessary information to calculate NDVI in the study area. This satellite was launched
into orbit by NASA on December 18, 1999. MODIS (Moderate Resolution Imaging
Spectroradiometer) sensor aboard this satellite collects information of different
reflectance bands. MODIS information is organized by "products". The product used in
this study was MOD09A1 (LP DAAC, 2014). MOD09A1 incorporates seven frequency
bands: Band 1 (620-670 nm), band 2 (841-876 nm), band 3 (459-479 nm), band 4
(545-565 nm), 5 band (1230-1250 nm), band 6 (1628-1652 nm) and band 7 (2105-2155





nm). The bands used to calculate NDVI are: band 1 for red frequency and band 2 for
near-infrared frequency. MOD09A1 provides georeferenced images with pixel
resolution of 500m x 500m. This product has a mix of the best reflectance measures of
each pixel in an 8-days period.

Daily data from the completed station of meteorological networks were utilized
during the period studied (2002 – 2017). Meteorological station is located in
40°41'46"N 3°45'54"W (elevation 1004 m a.s.l.), less than 2 km from the study area
(AEMET, 2017).

***2.3 Site description***
Six pixels (500m x 500m) are considered located in a pasture area at the north of
the Community of Madrid (Spain) between the municipalities of "Soto del Real" and
"Colmenar Viejo". The study area is located between meridians 3° 45' 00" and 3° 47'
00" W and parallels 40° 42' 00" and 40° 44' 00" N approximately (see Fig. 1).


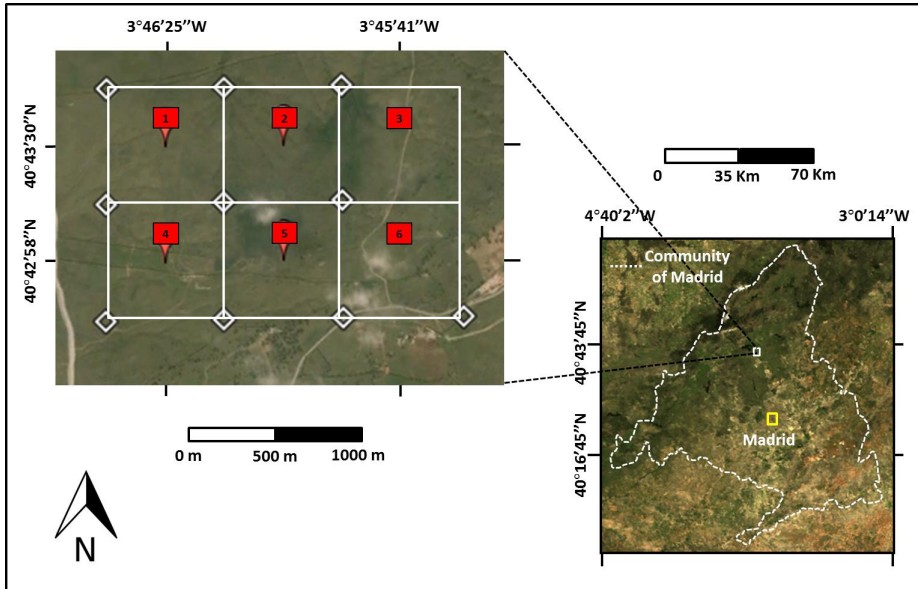


**Figure 1.** The study area is in the centre of the Iberian Peninsula (Community of Madrid). RGB
image of six pixels area used for case study is shown (Google Earth´s and MODIS images).

The annual mean temperature ranges during the study period from 12.7°C to
13.8°C, and annual mean precipitation ranges from 360 to 781 mm. The stations



studied were identified semi-arid (annual ratio P/ETo between 0.2 and 0.5) according
to the global aridity index developed by the United-Nations Convention to Combat
Desertification (UNEP, 1997). According to the climatic classification of Köppen (Kottek
et al., 2006), this area presents a continental Mediterranean climate temperate with
dry and temperate summer (type Csb). Temperature and precipitation of this site,
based on 20 years, is presented in Table 1.
Due to high soil moisture conditions, ash is the dominant tree, forming large
agroforestry systems ("dehesas") that are used for pasture. These are ecosystems with
high biodiversity.

**Table 1.** Monthly average of maximum temperature (Tmax), average temperature (Tavg) and
minimum temperature (Tmin) and precipitation (P).

| Month | Jan | Feb | Mar | Apr | May | Jun | Jul | Aug | Sep. | Oct | Nov | Dec | Annual |
|---|---|---|---|---|---|---|---|---|---|---|---|---|---|
| Tmax (ºC) | 7.1 | 9.3 | 12.7 | 15.4 | 19.5 | 24.6 | 28.6 | 28.1 | 23.7 | 16.8 | 11.1 | 7.4 | 17.0 |
| Tavg (ºC) | 3.6 | 4.8 | 7.7 | 10.1 | 13.7 | 18.4 | 22.0 | 21.7 | 17.9 | 12.3 | 7.1 | 4.1 | 12.0 |
| Tmin (ºC) | 0.0 | 0.3 | 2.6 | 4.8 | 7.8 | 12.1 | 15.4 | 15.3 | 12.0 | 7.8 | 3.0 | 0.8 | 6.8 |
| P (mm) | 67.2 | 50.0 | 38.5 | 62.2 | 62.3 | 30.2 | 18.9 | 16.4 | 34.2 | 79.3 | 86.2 | 82.6 | 627.9 |


### *2.4 HSL model*
There is no doubt that time-series of NDVI data from satellite sensors carry useful
information, which can be used for characterizing seasonal dynamics of vegetation
(Fensholt et al., 2012; Forkel et al., 2013). However, due to unfavorable atmospheric
conditions during the data acquisition, NDVI time-series curve often contains noise
(Motohka et al., 2011; Park, 2013). Although most of the NDVI data products are
temporally composited through maximum value compositing (MVC) method (Holben,
1986) to retain relatively cloud-free data, residual noise still exists in the data, which
will affect the accuracy of the NDVI value.
Therefore, usually it is necessary to reconstruct of NDVI time-series before
extracting information from the noisy data. There are several techniques that have
been applied to reduce noise and reconstruct NDVI series, a summary of these can be
found in Wei et al. (2016). In this study we applied a simple filtering method based on
the Hue-Saturation-Lightness (HSL) color model inspired by the work presented by
Tackenberd (2007).
HSL color model is a cylindrical representation of RGB (Red-Green-Blue) points.
Their components are Hue (color type), Saturation (level of color purity) and Lightness





(color luminosity). Hue is the angular component and it is more intuitive for humans
since it is directly related to the color wheel (see Fig. 2).

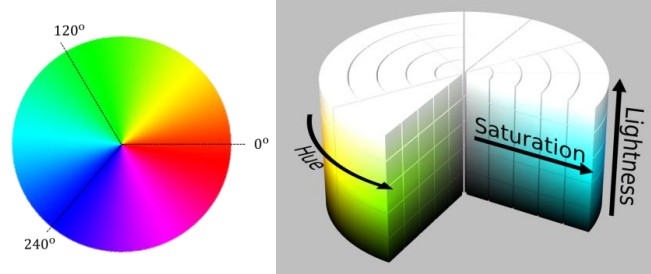

**Figure 2.** Colour wheel of Hue (on the left) and the HSL model (on the right).
Saturation is the radial component and near-zero values indicate grey colors.
Lightness is the axial radial versus axial component, zero lightness produces black and
full lightness produces white.
The NDVI series are filtered using the following HSL criteria: NDVI values are valid
if HSL Saturation is greater than 0.15. In this way, the values of the series that have
grey color correlated with pasture covered by clouds or snow are eliminated. This type
of filter based in HSL color space has been used on digital camera images monitoring
vegetation phenology (Tackenberg, 2007; Crimmins and Crimmins, 2008; Graham et
al., 2009). However, we have not found it in the context of remote sensing images.

### 2.5 Maximum Likelihood Method (MLM)
MLM estimates the set of parameters $\{\alpha, \beta, \mu, \sigma, \dots\}$ for a specific statistical
distribution that maximizes the "likelihood function" or the "joint density function":

$$L = f(\boldsymbol{x}, \boldsymbol{\theta}) = \prod_{i=1}^{n} f(x_i; \alpha, \beta, \mu, \sigma, \dots) \tag{3}$$

where $\boldsymbol{x} = (x_1, \dots, x_n)$ is the set of data, $\boldsymbol{\theta} = (\alpha, \beta, \mu, \sigma, \dots)$ is the vector of
parameters and $f(x_i; \alpha, \beta, \mu, \sigma, \dots)$ is the density function of the statistical model.
When maximization with respect to the vector of parameters is carried out, the
estimated parameters $(\hat{\alpha}, \hat{\beta}, \hat{\mu}, \hat{\sigma}, \dots)$ for the proposed statistical distribution are
obtained (Larson, 1982). Properties of estimated parameters are: invariance,
consistency and asymptotically unbiased.
In the case of a Gaussian model, the estimated statistics $\mu$ and $\sigma$ are defined by
accurate expressions as follows:




$$\hat{\mu} = \bar{x} = \frac{1}{n}\sum_{i=1}^{n} x_i \quad \hat{\sigma} = s = \sqrt{\frac{1}{n}\sum_{i=1}^{n}(x_i - \bar{x})^2} \tag{4}$$

where $\hat{\mu}$ is the sample mean and $\hat{\sigma}$ is the sample standard deviation of the data set.

### 2.6 Goodness of fit (Chi-square test)

$\chi^2$ test can be used to determine to what extent observed frequencies differ from frequencies expected for a specific statistical model. The most important points of the theory are briefly presented in (Cochran, 1952).

Let $f(x,\theta)$ be a theoretical density function of a random variable X which depends on parameters $\theta = (\alpha, \beta, \mu, \sigma, \dots)$ and let $x_1, \dots, x_n$ be a sample of X grouped into k classes with $n_i$ data per class i.

Firstly, the following hypothesis is set:

($H_0$) observed data fit theoretical distribution $f(x,\theta)$.

Then the test statistic $\chi_c^2$ is defined as:

$$\chi_c^2 = \sum_{i=0}^{k} \frac{(n_i - e_i)^2}{e_i} \tag{5}$$

where $n_i$ is the number of data or observed frequency and $e_i = n \cdot P(class\ i)$ is the expected frequency for class i. $P(class\ i)$ is the theoretical interval probability defined for class i.

A level of significance is also set as:

$$\alpha = P(Reject H_0 \ / \ H_0 is\ true\ ) \tag{6}$$

Finally, the following decision rule is applied: "reject the theoretical distribution at significance level $\alpha$ if:

$$\chi_c^2 = \sum_{i=0}^{k} \frac{(n_i - e_i)^2}{e_i} > \chi_{(k-m-1,1-\alpha)}^2 \tag{7}$$

where $\chi_{(k-m-1,1-\alpha)}^2$ is a $\chi^2$ distribution with k-m-1 degrees of freedom (m is the number of parameters, k is the number of classes).

## 3. Results and Discussion

### 3.1 HSL filtering criteria

NDVI series (from 2002 to 2017) were obtained for each pixel of the study area using frequency bands provided by MODIS product named MOD09A1. These series





contain some irregular values that can skew NDVI pattern. Therefore, the six series (six
pixels) were filtered using the HSL criteria. In Fig. 3 is shown an example of how HSL
filtering criteria works with a NDVI series.

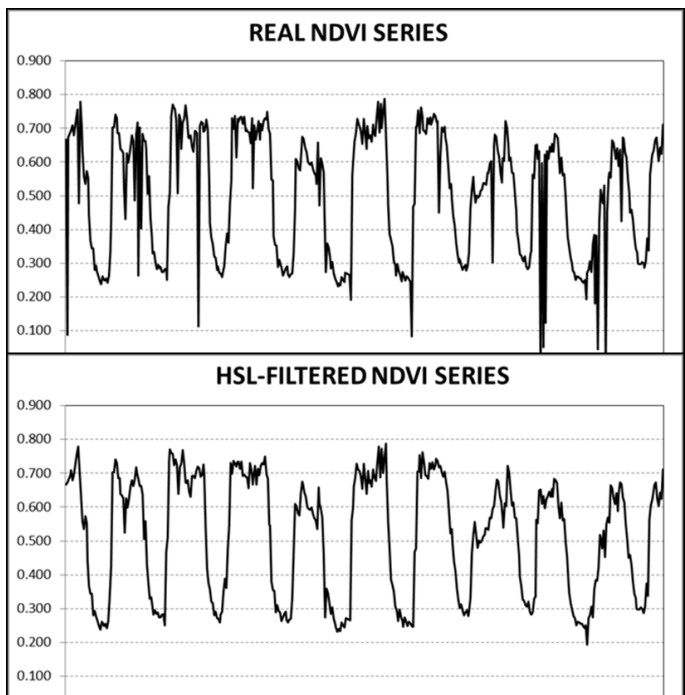


**Figure 3.** HSL filtering criteria applied to a NDVI series.

360        The abrupt changes in the NDVI values, mainly observed during raining seasons
such as autumn and winter, are efficiently eliminated. Not to be a high computational
demanding method is one of the main advantages of HSL filtering method. Therefore,
this method will allow us to obtain more robust NDVI values to be used in the
statistical analysis.

***3.2 Maximum Likelihood Method (MLM) and Chi square test***

367        In this study, a random variable (RV) of NDVI values has been defined every 8 days
(temporal resolution of MODIS product), in such a way that 46 RVs have been obtained
for the whole year. In Table 2, the definition of each RV can be seen, namely, the
period of the year (interval) which belongs to, and the amount of available NDVI
samples. Each RV collects the samples coming from the six selected pixels.






**Table 2.** Description of all RV defined in a year. Start - end of intervals and amount of samples
are shown.

| RANDOM VARIABLE | START PERIOD | END PERIOD | # SAMPLES | RANDOM VARIABLE | START PERIOD | END PERIOD | # SAMPLES |
|---|---|---|---|---|---|---|---|
| Interval 1 | 1-Jan | 8-Jan | 85 | Interval 24 | 4-Jul | 11-Jul | 96 |
| Interval 2 | 9-Jan | 16-Jan | 84 | Interval 25 | 12-Jul | 19-Jul | 96 |
| Interval 3 | 17-Jan | 24-Jan | 96 | Interval 26 | 20-Jul | 27-Jul | 96 |
| Interval 4 | 25-Jan | 1-Feb | 96 | Interval 27 | 28-Jul | 4-Aug | 96 |
| Interval 5 | 2-Feb | 9-Feb | 95 | Interval 28 | 5-Aug | 12-Aug | 96 |
| Interval 6 | 10-Feb | 17-Feb | 90 | Interval 29 | 13-Aug | 20-Aug | 96 |
| Interval 7 | 18-Feb | 25-Feb | 86 | Interval 30 | 21-Aug | 28-Aug | 96 |
| Interval 8 | 26-Feb | 5-Mar | 83 | Interval 31 | 29-Aug | 5-Sep | 96 |
| Interval 9 | 6-Mar | 13-Mar | 96 | Interval 32 | 6-Sep | 13-Sep | 96 |
| Interval 10 | 14-Mar | 21-Mar | 96 | Interval 33 | 14-Sep | 21-Sep | 94 |
| Interval 11 | 22-Mar | 29-Mar | 74 | Interval 34 | 22-Sep | 29-Sep | 96 |
| Interval 12 | 30-Mar | 6-Apr | 88 | Interval 35 | 30-Sep | 7-Oct | 96 |
| Interval 13 | 7-Apr | 14-Apr | 88 | Interval 36 | 8-Oct | 15-Oct | 85 |
| Interval 14 | 15-Apr | 22-Apr | 88 | Interval 37 | 16-Oct | 23-Oct | 90 |
| Interval 15 | 23-Apr | 30-Apr | 96 | Interval 38 | 24-Oct | 31-Oct | 96 |
| Interval 16 | 1-May | 8-May | 92 | Interval 39 | 1-Nov | 8-Nov | 92 |
| Interval 17 | 9-May | 16-May | 88 | Interval 40 | 9-Nov | 16-Nov | 90 |
| Interval 18 | 17-May | 24-May | 96 | Interval 41 | 17-Nov | 24-Nov | 96 |
| Interval 19 | 25-May | 1-Jun | 95 | Interval 42 | 25-Nov | 2-Dec | 89 |
| Interval 20 | 2-Jun | 9-Jun | 96 | Interval 43 | 3-Dec | 10-Dec | 95 |
| Interval 21 | 10-Jun | 17-Jun | 95 | Interval 44 | 11-Dec | 18-Dec | 88 |
| Interval 22 | 18-Jun | 25-Jun | 96 | Interval 45 | 19-Dec | 26-Dec | 90 |
| Interval 23 | 26-Jun | 3-Jul | 96 | Interval 46 | 27-Dec | 31-Dec | 90 |



In Fig. 4, a plot with NDVI sample means of all RV with a start and end reference of
the astronomical seasons is shown. The typical evolution of the NDVI along a year can
be seen.













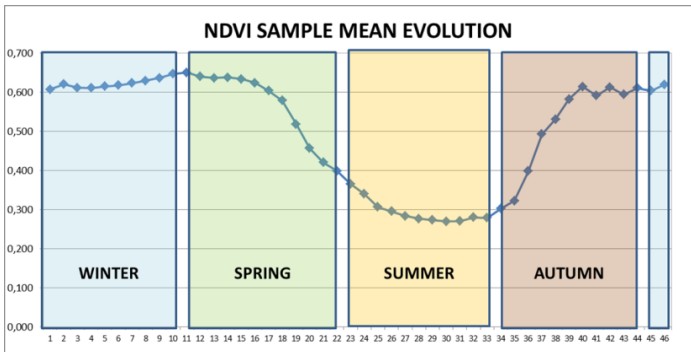


**Figure 4.** NDVI sample means of 46 random variables (RV) are shown as well as start and end
reference of every season.


The observed evolution of NDVI through the different seasons is typical of the
pasture in this area. The summer presents the lowest mean values which begin to
increase in autumn achieving a maximum mean value of 0.60 or 0.65 during winter. In
the middle of the spring NDVI decrease again, approaching the lowest mean value of
0.28 approximately.

Taking into account these values, dense vegetation, in this study pasture, is found
from middle of October (interval 37) till the end of May (interval 19) (see Table 2). It is
in this period where the precipitation concentrates (see Table 1). During the summer,
the NDVI mean values are lower than 0.3 corresponding with low precipitation and
high temperatures.

Following the work of Escribano-Rodriguez et al. (2014), there is a relationship of
pasture damage and a NDVI value around 0.40. Even if the authors point out that this
value is highly variable depending on the location, we can see that summer season in
this case study is under this value (see Fig. 4). This can explain that "Insurances for
Damaged Pasture" usually do not apply in these dates due to the arid environment
(BOE, 2013).

MLM has been applied to model these 46 RV. Parameters have been calculated for
4 probability density functions (PDF) which are the candidates to be the best fit. In
Table 3, a brief description is presented of these PDF candidates: Normal, Gamma,
Beta and Generalized Extreme Values (GEV). To do so, the following MATLAB functions
have been used: "normfit", "gamfit", "betafit" and "gevfit" (respectively).




**Table 3.** Candidate Probability Density Functions (PDF).

| PDF NAME | PDF EXPRESSION | PDF PARAMETERS |
|---|---|---|
| Normal | $f(x; \mu, \sigma) = \dfrac{1}{\sigma\sqrt{2\pi}} e^{-\frac{1}{2}\left(\frac{x-\mu}{\sigma}\right)^2}$ | $\mu \equiv average$ <br> $\sigma \equiv standard\ deviation$ |
| Gamma | $f(x; \alpha, \beta) = \dfrac{1}{\beta^\alpha \Gamma(\alpha)} x^{\alpha-1} e^{-\frac{x}{\beta}}$ | $\Gamma(.) \equiv gamma\ function$ <br> $\alpha\ and\ \beta \equiv parameters$ |
| Beta | $f(x; a, b) = \dfrac{\Gamma(a+b)}{\Gamma(a)\Gamma(a)} x^{a-1}(1-x)^{b-1}$ | $\Gamma(.) \equiv gamma\ function$ <br> $a\ and\ b \equiv parameters$ |
| Generalized Extreme Values (GEV) | $f(x; \mu, \sigma, \xi) = \dfrac{1}{\sigma} t(x)^{\xi+1} e^{-t(x)}$ <br><br> where $t(x) = \begin{cases} \left(1 + \left(\frac{x-\mu}{\sigma}\right)\xi\right)^{-1/\xi} & \text{if } \xi \neq 0 \\ e^{-(x-\mu)/\sigma} & \text{if } \xi = 0 \end{cases}$ | $\mu \in \mathbb{R} \equiv location\ param.$ <br> $\sigma > 0 \equiv scale\ parameter$ <br> $\xi \in \mathbb{R} \equiv shape\ parameter$ |


To check the goodness of the fit of PDF candidates, Chi square test ($\chi^2$ test) has
been used from 7 classes to 14 classes meeting the requirement that each class has at
least five observations. To calculate $\chi^2$, the theoretical probability defined for class i,
the following MATLAB functions have been used: "normcdf", "gamcdf", "betacdf" and
"gevcdf", which represent the cumulative density functions of each RV.

Twelve intervals (from 23 to 34) corresponding to months of July, August and
September have been excluded of this analysis since these intervals fall into the dry
season in the study area, normally not cover by any SIBI. Fig. 5 shows the percentage
of intervals that fit for every PDF candidate. The number of classes used in $\chi^2$ test is
represented at X-axis (from 7 to 14 classes).

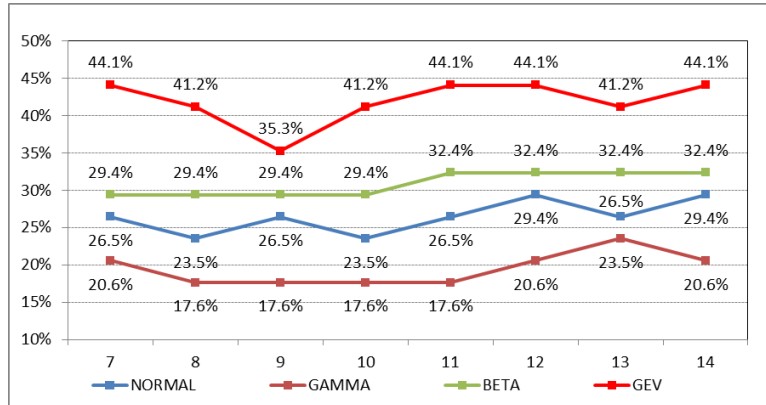


**Figure 5.** Percentage of fitted intervals for each PDF candidate (normal, gamma, beta and GEV
distributions).





Fig. 5 indicates that GEV distributions explain more intervals (more than 40% for
the majority of different class analysis) than normal, gamma or beta distributions.
Therefore, the methodology applied in Turvey et al. (2012) to design an index-based
insurance using NDVI values will not be feasible in this case study. This one uses an
average and standard deviation to define a percentage of cases where NDVI value will
be lower than a threshold defined by normal parameters. An important different
between the normal distribution and the rest of the PDF used in this work is its
symmetry and kurtosis. Many of the observed NDVI frequency distributions present a
clear asymmetry and long tails in one or both sides that causes normal distribution not
to be the optimal fit.
Table A1 at Appendix A shows the estimated parameters for each PDF and each
interval calculated by the MLM. These parameters will be used to compare the
estimated PDF with the NDVI observed values on different times through the seasons.
The following intervals are shown as examples of better GEV fit: interval 4 and 8 (for
winter, see Fig. 6), interval 17 and 21 (for spring, see Fig. 7) and interval 36 and 40 (for
autumn, see Fig. 8). In these plots, observed frequency is compared versus normal and
GEV density distributions calculated by MLM.

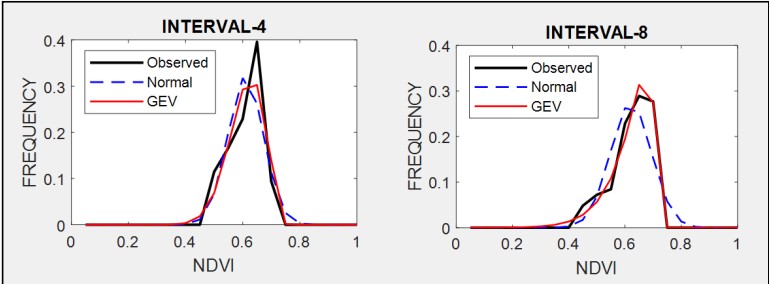


**Figure 6.** Comparison between observed NDVI frequency, GEV and normal probability density
functions (PDF) on two different dates. Intervals 4 and 8 are examples for winter.

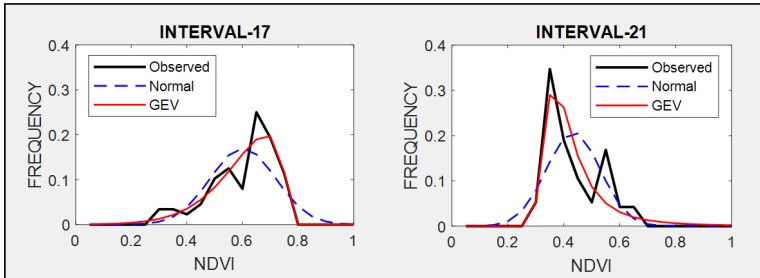


**Figure 7.** Comparison between observed NDVI frequency, GEV and Normal probability density
functions (PDF) on two different dates. Intervals 17 and 21 are examples for spring.






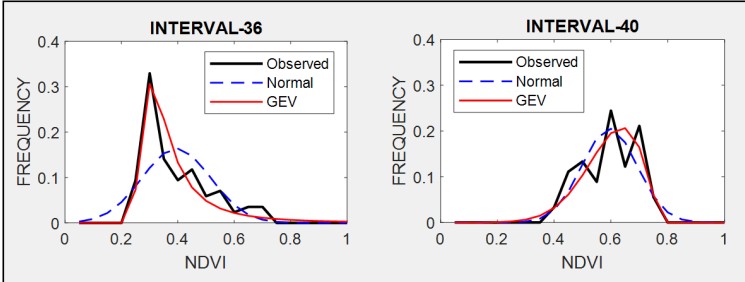


**Figure 8.** Comparison between observed NDVI frequency, GEV and normal probability density
functions (PDF) on two different times. Intervals 36 and 41 are examples for autumn.
During winter (see Fig. 6) the observed NDVI distribution presents negative
skewness. Then, there is a higher frequency of high NDVI values corresponding with
significant precipitation. During spring an evolution in the skewness is observed
passing from negative to positive, and so, the lower NDVI values become the higher
probable. Finally, during autumn precipitation begins and from positive pass to
negative skewness and higher NDVI values are possible. We can observe that normal
distribution has no flexibility to follow this dynamic in the distributions on each time.
This comparison is done in a sequential order for the whole of intervals in Figures A1,
A2, A3 and A4 at Appendix A.

The more skewness and kurtosis depart from those of the normal distribution the
larger the errors affecting the insurance designed based on (Turvey et al., 2012). It is
an expected result as pasture scenario is quite different from the development of a
crop, where normal distributions in the NDVI values are more expected. This high
heterogeneity in time and space of NDVI estimated on pasture has been pointed out in
several works (Martin-Sotoca et al, 2018). At the same time, more different is the
observed NDVI frequency from a normal distribution less representative is the
average, and so, the median becomes a more representative value.

***3.3 Insurance context***
The use of NDVI thresholds in damaged pasture context was presented in the
introduction section, being an example of using the "Insurance for Damaged Pasture"
in Spain. We have chosen this last insurance to compare the results between applying
Normal and GEV distribution methodologies. In this particular case the NDVI threshold
( $NDVI_{th}$ ) was calculated using the expression $NDVI_{th} = \mu - k \cdot \sigma$ (where $\mu, \sigma$ are
average and standard deviation of NDVI distributions respectively, assuming the
Normal hypothesis).






The probability of being below $NDVI_{th}$ (using $k = 0.7$, first damage level in the
insurance) at every interval has been calculated assuming the Normal hypothesis. As it
was expected, this value is always 24.2% (see third column in Table 4). The probability
of being below $NDVI_{th}$ has also been calculated using GEV distributions obtained in
this study. The probability obtained by GEV distributions is mostly lower than the
Normal distributions in spring, autumn and winter (see Table 4) that is the working
period of the insurance.

Observing where in time are localized the highest relative error in probabilities
(fifth column in Table 4), in absolute values, intervals corresponding to the end of
winter, second middle of spring and the beginning of autumn present errors higher
than 10%. This could explain why it is in spring and autumn when more disagreements
exist between farmers and insurance company in claims.

**Table 4 – First column:** time intervals of approximately 8 days along the year. **Second column:**
NDVI thresholds (*NDVI$_{th}$*) based on a Normal distribution applying $\mu - 0.7 \times \sigma$. **Third column:**
percentages of area below the *NDVI$_{th}$* when Normal distributions are applied. **Fourth column:**
percentages of area below the *NDVI$_{th}$* when GEV distributions are applied. **Fifth column:** relative
area error of GEV compared to the Normal distribution.

| RANDOM VARIABLE | NORMAL | | GEV | |
|---|---|---|---|---|
| | *NDVI$_{th}$* | Prob. | Prob. | Error (%) |
| Interval 1 | 0.535 | 24.20% | 24.37% | 0.70% |
| Interval 2 | 0.541 | 24.20% | 23.18% | -4.21% |
| Interval 3 | 0.541 | 24.20% | 23.27% | -3.84% |
| Interval 4 | 0.543 | 24.20% | 23.27% | -3.84% |
| Interval 5 | 0.545 | 24.20% | 24.17% | -0.12% |
| Interval 6 | 0.534 | 24.20% | 21.48% | -11.24% |
| Interval 7 | 0.528 | 24.20% | 24.01% | -0.79% |
| Interval 8 | 0.546 | 24.20% | 20.70% | -14.46% |
| Interval 9 | 0.555 | 24.20% | 21.30% | -11.98% |
| Interval 10 | 0.561 | 24.20% | 22.28% | -7.93% |
| Interval 11 | 0.567 | 24.20% | 23.49% | -2.93% |
| Interval 12 | 0.572 | 24.20% | 23.75% | -1.86% |
| Interval 13 | 0.571 | 24.20% | 23.20% | -4.13% |
| Interval 14 | 0.570 | 24.20% | 24.29% | 0.37% |
| Interval 15 | 0.571 | 24.20% | 23.47% | -3.02% |
| Interval 16 | 0.560 | 24.20% | 23.26% | -3.88% |





| | | | | |
|---|---|---|---|---|
| **Interval 17** | 0.495 | 24.20% | 21.29% | -12.02% |
| **Interval 18** | 0.484 | 24.20% | 21.58% | -10.83% |
| **Interval 19** | 0.442 | 24.20% | 23.06% | -4.71% |
| **Interval 20** | 0.381 | 24.20% | 27.20% | 12.40% |
| **Interval 21** | 0.342 | 24.20% | 29.46% | 21.74% |
| **Interval 22** | 0.323 | 24.20% | 28.84% | 19.17% |
| **Interval 35** | 0.257 | 24.20% | 18.98% | -21.57% |
| **Interval 36** | 0.285 | 24.20% | 28.57% | 18.06% |
| **Interval 37** | 0.333 | 24.20% | 25.90% | 7.02% |
| **Interval 38** | 0.398 | 24.20% | 24.27% | 0.29% |
| **Interval 39** | 0.454 | 24.20% | 23.79% | -1.69% |
| **Interval 40** | 0.503 | 24.20% | 22.81% | -5.74% |
| **Interval 41** | 0.491 | 24.20% | 23.23% | -4.01% |
| **Interval 42** | 0.517 | 24.20% | 24.66% | 1.90% |
| **Interval 43** | 0.507 | 24.20% | 23.13% | -4.42% |
| **Interval 44** | 0.514 | 24.20% | 23.49% | -2.93% |
| **Interval 45** | 0.515 | 24.20% | 23.70% | -2.07% |
| **Interval 46** | 0.509 | 24.20% | 23.33% | -3.60% |


In Table 4, Normal $NDVI_{th}$ have been used to calculate the probability in GEV
distributions. An alternative calculation can be the use of normal probability (24.2%) to
calculate new $NDVI_{th}$ based on GEV (see Table 5). It can be seen that new $NDVI_{th}$
obtained by GEV distributions are mostly upper than thresholds using Normal
distributions in spring, autumn and winter. Considering these results we find that
damage thresholds calculated by GEV methodology are mostly above that one's
calculated by Normal methodology.
Again, intervals corresponding to the end of winter, second middle of spring and the
beginning of autumn present $NDVI_{th}$ relative errors higher than 1% in absolute values
(fourth column in Table 5).

**Table 5 - First column:** time intervals of approximately 8 days along the year. **Second column:** NDVI
thresholds ($NDVI_{Th}$) based on a Normal distribution (Normal) applying $\mu - 0.7 \times \sigma$. **Third column:**
$NDVI_{Th}$ based on a GEV distribution (GEV) using 24.2% as the area below the $NDVI_{Th}$. **Fourth**
**column:** relative $NDVI_{Th}$ error of GEV compared to the Normal distribution.




| RANDOM VARIABLE | NDVI$_{Th}$ | | Error (%) |
|---|---|---|---|
| | Normal | GEV | |
| Interval 1 | 0.535 | 0.534 | -0,19% |
| Interval 2 | 0.541 | 0.543 | 0,37% |
| Interval 3 | 0.541 | 0.543 | 0,37% |
| Interval 4 | 0.543 | 0.545 | 0,37% |
| Interval 5 | 0.545 | 0.545 | 0,00% |
| Interval 6 | 0.534 | 0.543 | 1,69% |
| Interval 7 | 0.528 | 0.528 | 0,00% |
| Interval 8 | 0.546 | 0.558 | 2,20% |
| Interval 9 | 0.555 | 0.563 | 1,44% |
| Interval 10 | 0.561 | 0.567 | 1,07% |
| Interval 11 | 0.567 | 0.569 | 0,35% |
| Interval 12 | 0.572 | 0.574 | 0,35% |
| Interval 13 | 0.571 | 0.574 | 0,53% |
| Interval 14 | 0.570 | 0.569 | -0,18% |
| Interval 15 | 0.571 | 0.573 | 0,35% |
| Interval 16 | 0.560 | 0.563 | 0,54% |
| Interval 17 | 0.495 | 0.510 | 3,03% |
| Interval 18 | 0.484 | 0.498 | 2,89% |
| Interval 19 | 0.442 | 0.447 | 1,13% |
| Interval 20 | 0.381 | 0.374 | -1,84% |
| Interval 21 | 0.342 | 0.334 | -2,34% |
| Interval 22 | 0.323 | 0.318 | -1,55% |
| Interval 35 | 0.257 | 0.262 | 1,95% |
| Interval 36 | 0.285 | 0.278 | -2,46% |
| Interval 37 | 0.333 | 0.327 | -1,80% |
| Interval 38 | 0.398 | 0.398 | 0,00% |
| Interval 39 | 0.454 | 0.455 | 0,22% |
| Interval 40 | 0.503 | 0.508 | 0,99% |
| Interval 41 | 0.491 | 0.494 | 0,61% |
| Interval 42 | 0.517 | 0.516 | -0,19% |
| Interval 43 | 0.507 | 0.510 | 0,59% |
| Interval 44 | 0.514 | 0.516 | 0,39% |
| Interval 45 | 0.515 | 0.516 | 0,19% |
| Interval 46 | 0.509 | 0.511 | 0,39% |


## 4. Conclusions

According to the results obtained in the study area using MLM and $\chi^2$ test, it can
be concluded that normal distributions are not the best fit to the NDVI observations,
and GEV distributions provide a better approximation.



The difference between Normal and GEV assumption is more evident in the
transition from winter to summer (spring), where NDVI values decrease, and then from
summer to winter (autumn) presenting the opposite behavior of increasing NDVI
values. In both periods asymmetrical distributions were found, negative skewness for
the spring transition and positive skewness for the autumn transition. During both
periods the variability in precipitation and temperatures were higher in this location.
We have found differences if GEV assumption is selected instead of the Normal
one when defining damaged pasture thresholds ($NDVI_{th}$). The use of these different
assumptions should be taken into account in future insurance implementations due to
the important consequences of supposing a damage event or not. We propose the use
of percentiles in experimental NDVI distributions instead of average and standard
deviation, typically of normal distributions, to calculate new  $NDVI_{th}$.

## Acknowledgements

This research has been partially supported by funding from MINECO under contract
No. MTM2015-63914-P and CICYT PCIN-2014-080.



**Appendix A**

561            **Table A1 -** Maximum Likelihood parameters calculated for 4 PDF.

| RANDOM VARIABLE | NORMAL | | GAMMA | | BETA | | GEV | | |
|---|---|---|---|---|---|---|---|---|---|
| | μ | σ | α | β | a | b | μ | σ | ξ |
| Interval 1 | 0.591 | 0.081 | 53.31 | 0.011 | 21.45 | 14.82 | 0.563 | 0.080 | -0.297 |
| Interval 2 | 0.589 | 0.069 | 71.14 | 0.008 | 30.62 | 21.40 | 0.571 | 0.073 | -0.477 |
| Interval 3 | 0.583 | 0.060 | 94.15 | 0.006 | 39.56 | 28.34 | 0.567 | 0.063 | -0.457 |
| Interval 4 | 0.585 | 0.060 | 91.88 | 0.006 | 39.58 | 28.05 | 0.570 | 0.064 | -0.468 |
| Interval 5 | 0.588 | 0.061 | 93.92 | 0.006 | 38.83 | 27.25 | 0.568 | 0.061 | -0.340 |
| Interval 6 | 0.582 | 0.068 | 70.28 | 0.008 | 30.67 | 22.05 | 0.577 | 0.083 | -0.846 |
| Interval 7 | 0.584 | 0.080 | 52.52 | 0.011 | 22.16 | 15.82 | 0.559 | 0.082 | -0.366 |
| Interval 8 | 0.596 | 0.071 | 65.37 | 0.009 | 28.89 | 19.59 | 0.591 | 0.081 | -0.833 |
| Interval 9 | 0.601 | 0.066 | 76.02 | 0.008 | 34.31 | 22.84 | 0.590 | 0.070 | -0.652 |
| Interval 10 | 0.613 | 0.073 | 63.83 | 0.010 | 27.80 | 17.62 | 0.598 | 0.079 | -0.572 |
| Interval 11 | 0.621 | 0.078 | 58.72 | 0.011 | 24.33 | 14.86 | 0.600 | 0.083 | -0.451 |
| Interval 12 | 0.624 | 0.073 | 68.33 | 0.009 | 28.01 | 16.94 | 0.603 | 0.078 | -0.431 |
| Interval 13 | 0.624 | 0.075 | 66.22 | 0.009 | 26.23 | 15.85 | 0.604 | 0.080 | -0.476 |
| Interval 14 | 0.631 | 0.088 | 50.23 | 0.013 | 18.71 | 10.92 | 0.603 | 0.090 | -0.342 |
| Interval 15 | 0.630 | 0.084 | 53.60 | 0.012 | 21.17 | 12.45 | 0.607 | 0.089 | -0.448 |
| Interval 16 | 0.627 | 0.096 | 38.75 | 0.016 | 16.08 | 9.59 | 0.602 | 0.103 | -0.474 |
| Interval 17 | 0.577 | 0.117 | 20.47 | 0.028 | 10.24 | 7.58 | 0.560 | 0.127 | -0.692 |
| Interval 18 | 0.568 | 0.120 | 20.52 | 0.028 | 9.71 | 7.42 | 0.552 | 0.136 | -0.718 |
| Interval 19 | 0.523 | 0.116 | 19.46 | 0.027 | 9.52 | 8.68 | 0.495 | 0.125 | -0.493 |
| Interval 20 | 0.452 | 0.101 | 20.99 | 0.022 | 10.98 | 13.31 | 0.401 | 0.077 | 0.078 |
| Interval 21 | 0.409 | 0.095 | 19.94 | 0.021 | 11.18 | 16.13 | 0.354 | 0.060 | 0.325 |
| Interval 22 | 0.379 | 0.080 | 24.66 | 0.015 | 14.41 | 23.52 | 0.333 | 0.046 | 0.385 |
| Interval 23 | 0.353 | 0.073 | 26.54 | 0.013 | 15.85 | 29.01 | 0.311 | 0.036 | 0.456 |
| Interval 24 | 0.328 | 0.056 | 38.36 | 0.009 | 24.22 | 49.65 | 0.298 | 0.033 | 0.287 |
| Interval 25 | 0.305 | 0.044 | 53.52 | 0.006 | 35.62 | 81.20 | 0.282 | 0.028 | 0.210 |
| Interval 26 | 0.298 | 0.034 | 78.93 | 0.004 | 54.47 | 128.55 | 0.283 | 0.029 | -0.064 |
| Interval 27 | 0.289 | 0.026 | 126.85 | 0.002 | 88.33 | 217.15 | 0.278 | 0.021 | -0.030 |
| Interval 28 | 0.282 | 0.022 | 166.17 | 0.002 | 119.50 | 305.03 | 0.274 | 0.022 | -0.322 |
| Interval 29 | 0.278 | 0.021 | 179.09 | 0.002 | 127.93 | 332.63 | 0.269 | 0.018 | -0.085 |
| Interval 30 | 0.273 | 0.019 | 203.11 | 0.001 | 147.67 | 393.21 | 0.266 | 0.019 | -0.247 |
| Interval 31 | 0.272 | 0.022 | 166.83 | 0.002 | 120.11 | 321.95 | 0.262 | 0.018 | -0.059 |
| Interval 32 | 0.280 | 0.034 | 75.63 | 0.004 | 52.36 | 134.30 | 0.264 | 0.023 | 0.118 |
| Interval 33 | 0.285 | 0.034 | 82.05 | 0.004 | 54.90 | 137.68 | 0.270 | 0.020 | 0.122 |
| Interval 34 | 0.295 | 0.057 | 33.26 | 0.009 | 21.15 | 50.37 | 0.268 | 0.024 | 0.363 |
| Interval 35 | 0.312 | 0.079 | 19.70 | 0.016 | 11.83 | 25.94 | 0.275 | 0.038 | 0.300 |
| Interval 36 | 0.369 | 0.121 | 10.81 | 0.034 | 6.11 | 10.33 | 0.298 | 0.063 | 0.480 |
| Interval 37 | 0.432 | 0.141 | 9.45 | 0.046 | 5.21 | 6.81 | 0.370 | 0.120 | -0.080 |



| | | | | | | | | | |
|---|---|---|---|---|---|---|---|---|---|
| Interval 38 | 0.487 | 0.128 | 13.88 | 0.035 | 7.25 | 7.63 | 0.445 | 0.127 | -0.321 |
| Interval 39 | 0.529 | 0.107 | 23.56 | 0.022 | 11.39 | 10.16 | 0.497 | 0.110 | -0.390 |
| Interval 40 | 0.570 | 0.096 | 34.02 | 0.017 | 15.10 | 11.40 | 0.548 | 0.105 | -0.533 |
| Interval 41 | 0.554 | 0.090 | 36.42 | 0.015 | 16.90 | 13.64 | 0.531 | 0.096 | -0.471 |
| Interval 42 | 0.583 | 0.095 | 37.29 | 0.016 | 15.56 | 11.11 | 0.551 | 0.094 | -0.295 |
| Interval 43 | 0.574 | 0.097 | 34.27 | 0.017 | 14.93 | 11.07 | 0.550 | 0.103 | -0.482 |
| Interval 44 | 0.572 | 0.083 | 47.13 | 0.012 | 20.40 | 15.26 | 0.549 | 0.086 | -0.425 |
| Interval 45 | 0.576 | 0.088 | 42.59 | 0.014 | 18.17 | 13.36 | 0.550 | 0.090 | -0.396 |
| Interval 46 | 0.570 | 0.088 | 41.98 | 0.014 | 18.11 | 13.66 | 0.546 | 0.092 | -0.445 |



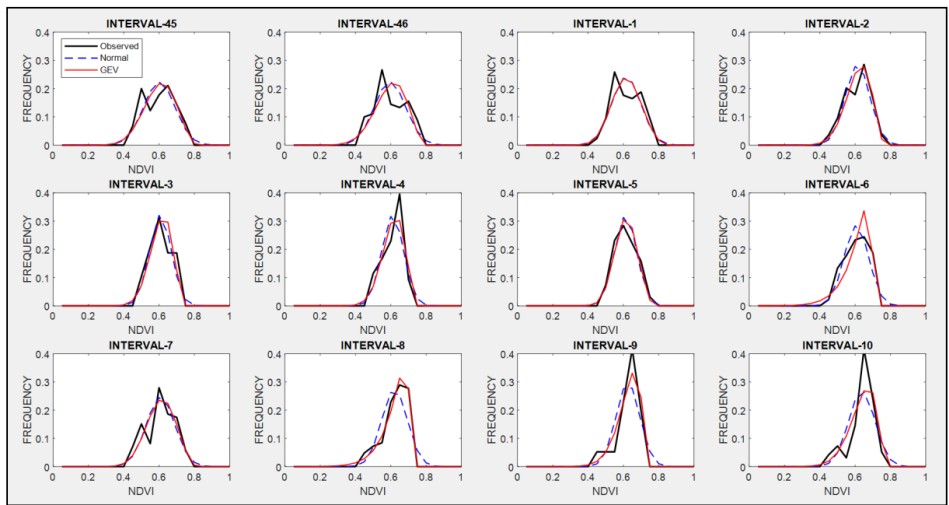


**Figure A1.** Observed NDVI, GEV and Normal probability density functions (PDF) from interval
45 to interval 10 (from 19 December to 21 March) representing Winter.





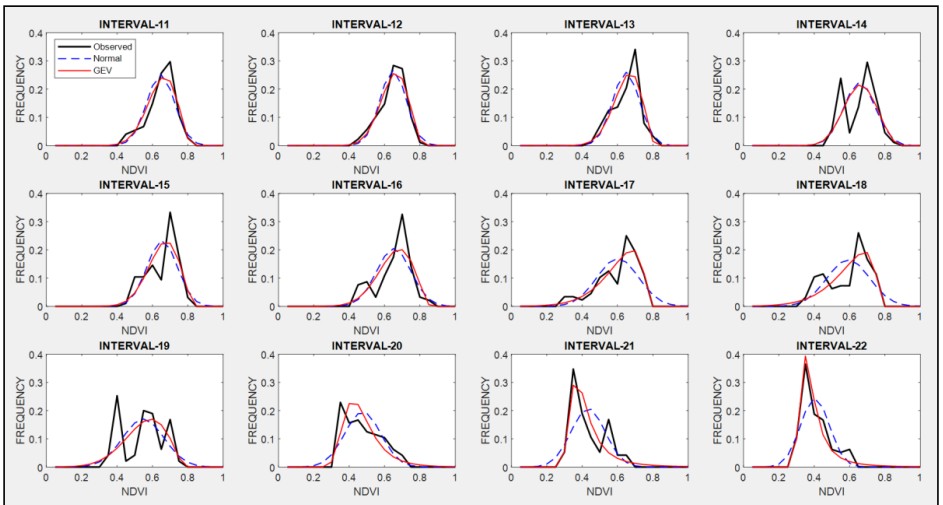


**Figure A2.** Observed NDVI, GEV and Normal probability density functions (PDF) from interval
11 to interval 22 (from 22 March to 25 June) representing Spring.

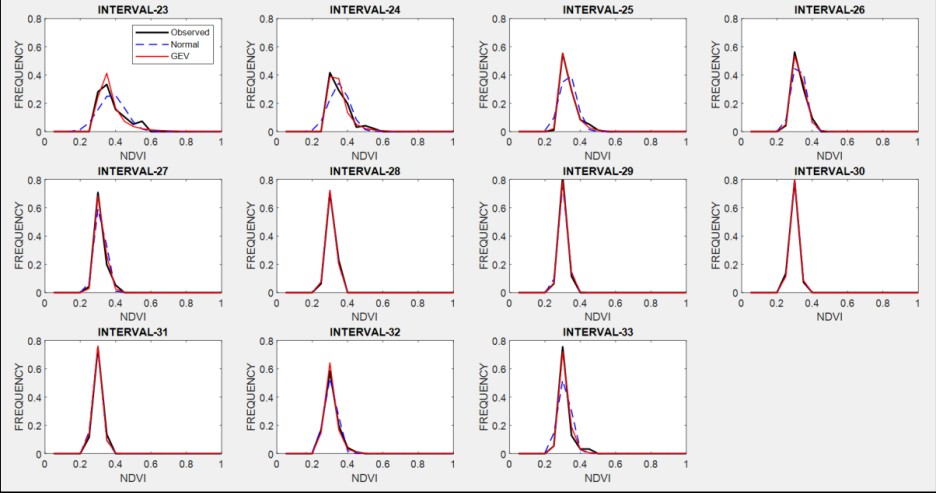


**Figure A3.** Observed NDVI, GEV and Normal probability density functions (PDFs) from interval
23 to interval 33 (from 26 June to 21 September) representing Summer.



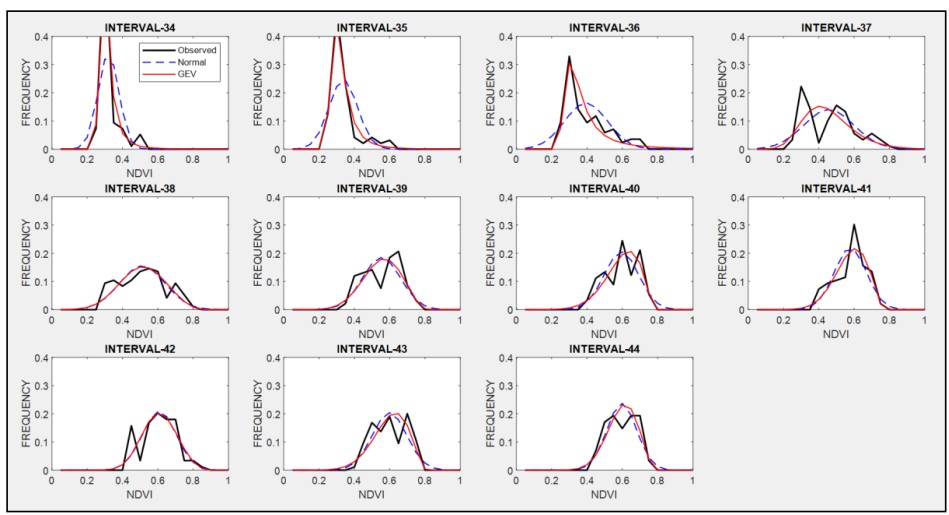

**Figure A4.** Observed NDVI, GEV and Normal PDFs from interval 34 to interval 44 (from 22 September to 18 December) representing Autumn.



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

Turvey, C.G.; Mcaurin, M.K.: Applicability of the Normalized Difference Vegetation
Index (NDVI) in Index-Based Crop Insurance Design, Am. Meorol. Soc., 4, 271-284,
681      2012.
UNEP Word Atlas of Desertification. Second Ed. United Nations Enviroment Programe,
Nairobi, 1997.
USDA. U.S. Department of Agriculture, Federal Crop Insurance Corporation, Risk
Management Agency: Rainfall Index Plan Annual Forage Crop Provisions. 16- RI-AF.
http://www.rma.usda.gov/policies/ri-vi/2015/16riaf.pdf 2013 (Accessed March 1,
687      2018).
Wei, W.; Wu, W.; Li,Z.; Yang, P.; Qingbo Zhou, Q.: Selecting the Optimal NDVI
Time-Series Reconstruction Technique for Crop Phenology Detection, Intell. Autom.
Soft. Co. 22, 237-247, 2016.