# Peer review of "Statistical Analysis for Satellite Index-Based Insurance to 1 define Damaged Pasture Thresholds 2"

_Natural Hazards and Earth System Sciences, 2019_

## Referee Comment (RC1) · Anonymous Referee #1 · 18 Mar 2019

1. GENERAL COMMENTS The article is very organized and structured. Making reading easier The language used is quite appropriate and the graphs and tables help to interpret the results so that none of them is left over The result that follows from this article seems very reasonable to me. In general, the use of percentiles to set damage thresholds is a more reliable method than the use of the mean and standard deviation, which would only be justified if the distribution of the index followed a Normal law. The authors show how the assumption of Normality is not very reasonable In addition, the article highlights the ability of insurance based on indices to verify the effects of crop or livestock losses, except table 2 that I think is redundant since it is known that the intervals go from 8 to 8 days and it is not clearly specified what is #samples, which

apparently is the most relevant of the table

**2. SPECIFIC COMMENTS**

In this study authors applied a simple filtering method based on the Hue-Saturation-Lightness (HSL). It is necessary to determine if there is any reason to explain why they apply this type of filter to reduce noise and not others.

There is information that is not clear although I suppose that the number of samples comes out of using 16 years and every day of the year a series, that would total 96 series for each period of 8 days in which the year has been divided. I suppose that some intervals have less data (series) because the atmospheric conditions of some days have prevented to obtain the complete sample. But in each sample, how many observations are we talking about? Or, instead of samples, was it meant to indicate observations? This point is important to calibrate the chi-squared test that is very sensitive to the number of observations of the sample to which you want to adjust. (There are other contrasts of goodness of fit that depending on the sample size could be more justified).

It is not clear that when checking the goodness of fit of each distribution which is the threshold p-value ($\alpha$) that has been chosen to consider that the RV variables follow the theoretical distributions that are being considered in each period.

Figure 5 shows the percentage of adjusted intervals for each candidate distribution. Is that correct? Is there any relationship between the season and the number of intervals that fit correctly for each type of distribution? That is not mentioned in the article. Are the authors satisfied with the results? In other words, the proportion of times that can be correctly adjusted to a type of distribution seems appropriate. What is the proportion from which you consider that percentage is satisfactory? It seems that what they want to present mainly is that the Normal distribution is not the one that best fits. When establishing the GEV distribution as an alternative, they have not statistically evaluated the differences between a GEV distribution and other triparamétricas (Generalized Pareto, Normal Log, Generalized Logistics, ...). I suppose that the justification may be due to the fact that the final solution they recommend is that it is a quantile of the RV that determines whether there is a drought or not.

In the results I am struck by the difference obtained in the parameters of the GEV distribution between close intervals, for example between 35 and 36 in the first the probability of obtaining a value below 0.257 is 0.1898, while in the second is 0.2857. Can you explain that difference.

3. TECHNICAL CORRECTIONS

âǍć In figure 3 it is necessary to define the axis of abscissa, it is assumed that it is the date, varying in the 16 years that are collected from the information (January 2002 - december 2017), but it is not specified. âǍć Table 2 should be reduced, only the number of samples per interval should be specified and clarify in the text that the intervals go consecutively from 8 to 8 days, indicating the start intervals of each season (winter, spring, autumn).

Please also note the supplement to this comment:
https://www.nat-hazards-earth-syst-sci-discuss.net/nhess-2019-34/nhess-2019-34-RC1-supplement.pdf

---

## Referee Comment (RC2) · Anonymous Referee #2 · 25 Mar 2019

General comments: This paper deals with the topic of agricultural insurance. It is focused on satellite index-based insurance, proposing an improvement in the insurance of damaged pasture based on NDVI distribution. The research carried out in this paper is of scientific and practical interest, and it is adequate to NHESS journal. The manuscript is well written, structured and presented. The methodology employed and the obtained results are clearly exposed. It seems to me that the paper could be published as long as the authors answer the following minor concerns that arose from the review process that I made.

Specific comments: I only have a couple of minor specific comments for the authors.

[Figure]

1. With respect to the differences found between the use of Normal distributions and GEV distributions (Figure 5), could authors affirm that such differences are statistically significant? 2. Did the authors apply this methodology on geographic areas of different characteristics with respect to the characteristics of the areas analyzed in the present study? Perhaps it would be necessary validating the robustness of the statistical method on different types of vegetation.

Minor comments: Page 1, line 21: The term "normal distribution" appears sometimes in lowercase and sometimes in uppercase. It would be desirable to homogenize this term throughout the entire manuscript. Page 1, line 22: Insert "Moderate Resolution Imaging Spectroradiometer" before "(MODIS)". This is the first time that it appears in the manuscript. Page 4, line 151: The meaning of MODIS must be deleted from here and included on Line 22. Page 6, line 235: The meaning of MODIS must be also deleted from here. Page 7, line 263: Add "mm" after "360". Page 8, line 275: Delete "and" and insert ",". Page 8, Table 1: In the first row of the table, delete the dot in "Sep.". Page 10, line 346: In the equation (7) it is not necessary to include the meaning of $\chi c2$, because it is just clear from equation (5). Page 11, line 358: For a better reader comprehension, authors could include the name (and scale) of the variables of both axes in graphs of Figure 3. Page 14, line 435: For the graph of Figure 5, similar comment to the previous one. Page 25, line 580: References should be adjusted to the format of the journal.

---

## Referee Comment (RC3) · Anonymous Referee #3 · 26 Mar 2019

This paper addresses the current use of NDVI values to assess insurances in agriculture and proposes a different approach to select the threshold values that relate to crop damage. The use of distribution functions other than the normal distribution is analysed on a NDVI time series obtained from 6-MODIS pixel sample (500 m x 500 m) in a study site in central Spain during 2002-2017. The topic is relevant due to the increasing occurrence of extreme conditions of drought on different time scales, and the need to monitor damage stages in a efficient way for both the insurance companies and the clients. The paper is generally well written and clearly presented, and the results show how the normal distribution assumption does not always hold. However, there are some limitations to recommend publication in its current version related to

some clarification to be done. English usage is generally good, although some final checking is recommended.

Specific comments:

1. A large fraction of the introduction is devoted to present how insurances based on indexes are being used and what constraints the indexes may have to accurately trace the damage level. However, the core (as the objective states) of the work is characterizing NDVI distribution functions; the introduction should also include a review of different works done on this, no matter for what application this was done, and what functions better succeed in reproducing the statistical behaviour of this variable on different time scales.

2. Additionally, the introduction should also refer to what limitations assuming the normal distribution has for NDVI characterization.

3. This being said, I think the objective should be more specific. On what time and spatial scales NDVI is defined for the work to be done? The variable should be very precisely defined.

4. I wonder how representative the presented study case is for the generalization of the results and conclusion. Different decisions made to develop this work should be justified: crop to evaluate, location, number of pixels, sensor... Why only some pixels and not the whole crop area?

5. The choice of the candidate functions (see Table 3) must also be justified.

6. Lines 441-444. This assertion is highly dependent on how representative the studied sample is of the case referred to, and to the general problem the paper wants to address.

7. From the figures in the Annex it is not so clear that GEV has an overall better performance than the normal choice. In fact, in different examples both perform very similarly.

[Figure]

8. I was expecting to find in the results more depth regarding the impact that a different distribution has on the parts (insurance companies and clients). At the end, even if statistically another function performs better than the normal assumption, the relevant issue is how much the benefit/loss is changed by this. A better performance might have only a minor impact on the final result in the insurance context. At least some estimation should be included.

9. Additionally, no discussion is done on other works in the results regarding the insurance context.

Minor comments:

10. Please, check that captions of tables and figures are self-explanatory (see for instance Table1; provide study period)

11. The weather variables statistics included in Table 1 just presents the local climate. Is it possible to combine this information with the NDVI monitoring to obtain better indexes or to validate the NDVI results?

12. Line 246. "...the completed station of meteorological networks". What do you mean by that?

13. Lines 306-311. What is the purpose of the last sentence?

14. Line 367. Please, define this more precisely.

15. Line 371. I think the definition of VR can be more clearly expressed. Additionally, the use of tables for VR intervals results in too long and repeated content.

16. Contents in lines 367-376 and 415-432 should be moved to the Methods sections.

17. Line 451. I would suggest not to use the future tense here.

18. Please, check the references' format meets the Journal's standards.

I hope the Authors find these comments useful and that they contribute to improve the

impact of the work done.

---

## Author Comment (AC1) · 16 Apr 2019

**TO REFEREE #1**

Thank you very much for all your suggestions and comments. Next, we respond all your suggestions in order.

1.  About Table 2:
    We have modified the table 2 eliminating redundant information.
2.  About HSL filtering method:
    We establish a relationship between the color of the satellite image and the color of the pasture contained in this image. Saturations lower than 0.15 are inconsistent with dry (low NDVI values) or healthy (high NDVI values) pasture and highly correlated with pasture covered by clouds or snow. Thus, this method uses a color criterion to eliminate wrong NDVI values.
3.  About the number of observations of every RV (interval):
    The theoretical number of observations for every RV is: 6 pixels x 16 year = 96 observations. We have lost some observations after applying the HSL filtering method. We have modified the word "sample" by "observations" to avoid misunderstanding.
4.  About the level of significance:
    You are right, we missed this important value. We have included it in the results. Now you can read: "The level of significance ($\alpha$) was fixed to 5% for all the candidates".
5.  About Figure 5:
    You are right, Fig. 5 shows the percentage of adjusted intervals (RVs) for each candidate distribution. We have added more information in the figure caption. Now you can read: "Figure 5. Percentage of fitted intervals (Y axis) for each PDF candidate (Normal, Gamma, Beta and GEV distributions) in function of the number of classes (X axis)."
6.  Is there any relationship between the season and the number of intervals that fit correctly for each type of distribution?
    When we filter the data by season we find that GEV distributions explain better some intervals of spring and autumn since their observed distributions are very asymmetric. On the other hand, we do not find an important difference in winter, since its observed distributions are mainly symmetric in these intervals.
7.  What is the proportion from which you consider that percentage is satisfactory?.
    In this study we do not want to affirm that GEV is the best distribution because fits better than the others. Our objective is to notice that could exist others alternatives to Normal distributions. With respect the selected distributions in this study we can affirm that 40% (GEV distribution) is highly enough to at least not
consider the Normal distribution.

8. "… you have not statistically evaluated the differences between GEV distribution
and other tri-parametric distributions (Generalized Pareto, Normal Log, …)":
The objective of this study is not to find the best fit for the observed NDVI
distribution, but to highlight that Normal distribution could not be the best fit. To
avoid this imprecision we recommend the use of quantiles to calculate damage
pasture thresholds.

9. Differences between interval 35 and 36:
These two intervals belong to autumn and this season is characterized by its high
variability. If you observe the NDVI distributions in the appendix A for these two
intervals, you can notice how the distribution is changing from summer (with a
strong peak) to autumn (with an incipient tail).

10. In figure 3 it is necessary to define the axis of abscissa:
Now you can see this information in the figure.

11. Clarify in the text that intervals go consecutively from 8 to 8 days, indicating the
start intervals of each season:
We have modified the first paragraph of section 3.2. Now you can read: "NDVI
values were obtained consecutively every 8 days from MODIS product starting at
1st of January of every year, in such a way that 46 NDVI observations were
considered for each year. Therefore, 46 Random Variables (RV) were defined when
taking into account all the years of this study.

[revised manuscript text omitted]

for each year. Therefore, 46 Random Variables (RV) were defined when taking into
account all the years of this study.

In Table 3, every RV (named as "Interval") can be seen together with the number of
available NDVI observations. Each RV collects the observations coming from the six
selected pixels. The start intervals of each season are: interval 45 for winter, interval 11
for spring, interval 23 for summer and interval 34 for autumn.

**Table 3.** Number of observations for every RV (named as Interval).

| RANDOM VARIABLE | # OBSERVATIONS | RANDOM VARIABLE | # OBSERVATIONS |
|---|---|---|---|
| Interval 1 | 85 | Interval 24 | 96 |
| Interval 2 | 84 | Interval 25 | 96 |
| Interval 3 | 96 | Interval 26 | 96 |
| Interval 4 | 96 | Interval 27 | 96 |
| Interval 5 | 95 | Interval 28 | 96 |
| Interval 6 | 90 | Interval 29 | 96 |
| Interval 7 | 86 | Interval 30 | 96 |
| Interval 8 | 83 | Interval 31 | 96 |
| Interval 9 | 96 | Interval 32 | 96 |
| Interval 10 | 96 | Interval 33 | 94 |
| Interval 11 | 74 | Interval 34 | 96 |
| Interval 12 | 88 | Interval 35 | 96 |
| Interval 13 | 88 | Interval 36 | 85 |
| Interval 14 | 88 | Interval 37 | 90 |
| Interval 15 | 96 | Interval 38 | 96 |
| Interval 16 | 92 | Interval 39 | 92 |
| Interval 17 | 88 | Interval 40 | 90 |
| Interval 18 | 96 | Interval 41 | 96 |
| Interval 19 | 95 | Interval 42 | 89 |
| Interval 20 | 96 | Interval 43 | 95 |
| Interval 21 | 95 | Interval 44 | 88 |
| Interval 22 | 96 | Interval 45 | 90 |
| Interval 23 | 96 | Interval 46 | 90 |

In Fig. 4, a plot with NDVI sample means of all RV with a start and end reference of
the astronomical seasons is shown. The typical evolution of the NDVI along a year can be
seen.

[Figure]

**Figure 4.** NDVI sample means of 46 random variables (RV) are shown as well as start and end
reference of every season. Study period from 2002 to 2017.

The observed evolution of NDVI through the different seasons is typical of the pasture
in this area. The summer presents the lowest mean values which begin to increase in
autumn achieving a maximum mean value of 0.60 or 0.65 during winter. In the middle of
the spring NDVI decrease again, approaching the lowest mean value of 0.28
approximately.

Taking into account these values, dense vegetation, in this study pasture, is found
from middle of October (interval 37) till the end of May (interval 19). It is in this period
where the precipitation concentrates (see Table 1). During the summer, the NDVI mean
values are lower than 0.3 corresponding with low precipitation and high temperatures.

Following the work of Escribano-Rodriguez et al. (2014), there is a relationship of
pasture damage and a NDVI value around 0.40. Even if the authors point out that this
value is highly variable depending on the location, we can see that summer season in this
case study is under this value (see Fig. 4). This can explain that "Insurances for Damaged
Pasture" usually do not apply in these dates due to the arid environment (BOE, 2013).

MLM has been applied to model these 46 RV. Parameters have been calculated for 4
PDF (see Table 2) which are the candidates to be the best fit. To check the goodness of the fit of PDF candidates, Chi square test ($\chi^2$ test) has been used from 7 classes to 14 classes
meeting the requirement that each class has at least five observations. The level of
significance ($\alpha$) was fixed to 5% for all the candidates.

Twelve intervals (from 23 to 34) corresponding to months of July, August and
September have been excluded of this analysis since these intervals fall into the dry
season in the study area, normally not cover by any SIBI. Therefore, calculations were
carried out over 34 intervals. Fig. 5 shows the percentage of intervals that fit for every PDF
candidate. The number of classes used in $\chi^2$ test is represented at X-axis (from 7 to 14
classes).

[Figure]

**Figure 5.** Percentage of fitted intervals (Y axis) for each PDF candidate (Normal, Gamma, Beta and
GEV distributions) in function of the number of classes (X axis).

Fig. 5 indicates that GEV distributions explain more intervals (more than 40% for the
majority of the class analysis) than Normal, Gamma or Beta distributions. An important
difference between the Normal distribution and the rest of the PDF used in this work is its
symmetry and kurtosis. Many of the observed NDVI distributions present a clear
asymmetry and long tails in one or both sides that causes Normal distribution not to be
the optimal fit.

There is a relationship between seasons and the number of intervals that fit correctly.
We found that GEV distributions explain better some intervals of spring and autumn since
their observed distributions are very asymmetric. On the other hand, we did not find an
important difference in winter, since its observed distributions are mainly symmetric.

Therefore, the methodology using the NDVI Normal assumption applied to design an
index-based insurance will not be feasible in many intervals of this study.

[revised manuscript text omitted]

22, 237-247, 2016.

---

## Author Comment (AC2) · 16 Apr 2019

**TO REFEREE #2**

Thank you very much for all your suggestions and comments. Next, we respond all your
suggestions in order.

1. With respect to the differences found between the use of Normal distributions and
GEV distributions, could authors affirm that such differences are statistically
significant?:

We recognize the limitations of this study in the selected time period. We were
limited to work with 16 years and approximately 96 observations per RV. However,
we have found that observed NDVI distributions are mainly asymmetric in spring
and autumn, inconsistent with symmetric Normal fitting.

In addition, we have included that the level of significance of the Chi-Square fit was
fixed to 5% for all the candidates (this information did not appear in the first
version and it will be included now).

Anyway, the objective of this study is to generate some reasonable doubts about
the convenience of using Normal distributions in all cases, and to notice that
others alternatives to Normal distributions could exist. GEV distribution is an
example of better fit than Normal one with the limitations explained above.

2. Did the authors apply this methodology on geographic areas of different
characteristics with respect to the characteristics of the area analyzed in the
present study?

In this study we have only focused on pasture and methodology applicable to
calculate damaged pasture thresholds. However, we also think this methodology
could be extrapolated to other types of vegetation in further researches.

3. Minor comments:

a) We have homogenized the term “Normal distribution” to uppercase.
b) Page 1, line 22: We have inserted Moderate Resolution Imaging
Spectroradiometer before MODIS in the Abstract.
c) Page 4, line 151: We have deleted the definition of MODIS.
d) Page 6, line 235: We have deleted the definition of MODIS.
e) Page 7, line 263: We have added mm after 360.
f) Page 8, line 275: We have inserted “.”.
g) Page 8, Table 1: We have deleted the dot.
h) Page 10, line 346: We have modified the equation (7).
i) Page 11, line 358: We have modified the graphs and included the scale and
name of axis.
j) Page 14, line 435: We have also modified this graph and the figure caption.
k) We have reviewed the format references.

**Statistical Analysis for Satellite Index-Based Insurance to**
**define Damaged Pasture Thresholds**

**Juan José Martín-Sotoca1\*, Antonio Saa-Requejo2,3, Rubén Moratiel2,3, Nicolas Dalezios4, Ioannis Faraslis5,**
**and Ana María Tarquis2,6**

[revised manuscript text omitted]
$\sigma \equiv \text{standard deviation}$                                                                               |
| Gamma  | $f(x; \alpha, \beta) = \frac{1}{\beta^\alpha \Gamma(\alpha)} x^{\alpha-1} e^{-\frac{x}{\beta}}$                                                                                                                                                   | $\Gamma(\cdot) \equiv \text{gamma function}$
$\alpha \text{ and } \beta \equiv \text{parameters}$                                                   |
| Beta   | $f(x; a, b) = \frac{\Gamma(a+b)}{\Gamma(a)\Gamma(b)} x^{a-1} (1-x)^{b-1}$                                                                                                                                                                         | $\Gamma(\cdot) \equiv \text{gamma function}$
$a \text{ and } b \equiv \text{parameters}$                                                            |
| GEV    | $f(x; \mu, \sigma, \xi) = \frac{1}{\sigma} t(x)^{\xi+1} e^{-t(x)}$
where $t(x) = \begin{cases} \left(1 + \left(\frac{x-\mu}{\sigma}\right)\xi\right)^{-1/\xi} & \text{if } \xi \neq 0 \\ e^{-(x-\mu)/\sigma} & \text{if } \xi = 0 \end{cases}$ | $\mu \in \mathbb{R} \equiv \text{location param.}$
$\sigma > 0 \equiv \text{scale parameter}$
$\xi \in \mathbb{R} \equiv \text{shape parameter}$ |

**2.6 Goodness of fit (Chi-square test)**

     $\chi^2$  test can be used to determine to what extent observed frequencies differ from
    frequencies expected for a specific statistical model. The most important points of the
    theory are briefly presented in (Cochran, 1952).

    Let  $f(x, \theta)$  be a theoretical density function of a random variable  $X$  which depends on
    parameters  $\theta = (\alpha, \beta, \mu, \sigma, \dots)$  and let  $x_1, \dots, x_n$  be a sample of  $X$  grouped into  $k$  classes with  $n_i$
    data per class  $i$ .

    Firstly, the following hypothesis is set:

                     $(H_0)$  observed data fit theoretical distribution  $f(x, \theta)$ .

    Then the test statistic  $\chi^2_c$  is defined as:

$$\quad \chi^2_c = \sum_{i=0}^k \frac{(n_i - e_i)^2}{e_i} \quad (5)$$

    where  $n_i$  is the number of data or observed frequency and  $e_i = n \cdot P(\text{class } i)$  is the
    expected frequency for class  $i$ .  $P(\text{class } i)$  is the theoretical interval probability defined for
    class  $i$ .

    A level of significance is also set as:

$$\quad \alpha = P(\text{Reject } H_0 / H_0 \text{ is true}) \quad (6)$$

    Finally, the following decision rule is applied: "reject the theoretical distribution at
    significance level  $\alpha$  if:

$$\quad \chi^2_c > \chi^2_{(k-m-1, 1-\alpha)} \quad (7)$$

where  $\chi^2_{(k-m-1, 1-\alpha)}$  is a  $\chi^2$  distribution with  $k-m-1$  degrees of freedom (m is the number of
parameters, k is the number of classes).

**385 **3. Results and Discussion**

**386 ***3.1 HSL filtering criterion**

NDVI series (from 2002 to 2017) were obtained for each pixel of the study area using
frequency bands provided by MODIS product named MOD09A1. These series contain
some irregular values that can skew NDVI pattern. Therefore, the six series (six pixels)
were filtered using the HSL criterion. **In Fig. 3 is shown an example of how HSL filtering**
**criterion works with a 10 years NDVI series (from 2002 to 2012).**

**Figure 3.** HSL filtering criterion applied to a 10 years NDVI series. Top graph shows the real NDVI
series. Bottom graph shows the HSL filtered NDVI series.

The abrupt changes in the NDVI values, mainly observed during raining seasons such
as autumn and winter, are efficiently eliminated. Not to be a high computational
demanding method is one of the main advantages of HSL filtering method. Therefore, this
method will allow us to obtain more robust NDVI values to be used in the statistical
analysis.

**3.2 Maximum Likelihood Method (MLM) and Chi square test**

       NDVI values were obtained consecutively every 8 days from MODIS product starting
       at 1st of January of every year, in such a way that 46 NDVI observations were considered
       for each year. Therefore, 46 Random Variables (RV) were defined when taking into
       account all the years of this study.

       In Table 3, every RV (named as “Interval”) can be seen together with the number of
       available NDVI observations. Each RV collects the observations coming from the six
       selected pixels. The start intervals of each season are: interval 45 for winter, interval 11
       for spring, interval 23 for summer and interval 34 for autumn.

       **Table 3.** Number of observations for every RV (named as Interval).

[revised manuscript text omitted]

There is a relationship between seasons and the number of intervals that fit correctly.
We found that GEV distributions explain better some intervals of spring and autumn since
their observed distributions are very asymmetric. On the other hand, we did not find an
important difference in winter, since its observed distributions are mainly symmetric.

Therefore, the methodology using the NDVI Normal assumption applied to design an
index-based insurance will not be feasible in many intervals of this study.

[revised manuscript text omitted]

22, 237-247, 2016.

---

## Author Comment (AC3) · 16 Apr 2019

**TO REFEREE #3**

Thank you very much for all your suggestions and comments. Next, we respond all your suggestions in order.

1. The core (as the objective states) of the work is characterizing NDVI distribution functions; the introduction should also include a review of different works done on this, no matter for what application this was done, and what functions better succeed in reproducing the statistical behaviour of this variable on different time scales.

   We present this study as a novelty of NDVI time characterization without Normal assumption. In the introduction we have explained NDVI characterization when it is used in the index-based insurance context, and assuming normal distributions.

2. Additionally, the introduction should also refer to what limitations assuming the normal distribution has for NDVI characterization.

   We have dealt with this topic in the result section because limitations assuming the normal distribution are part of our results and conclusions. Different NDVI distribution assumptions involve different damaged NDVI thresholds.

3. This being said, I think the objective should be more specific. On what time and spatial scales NDVI is defined for the work to be done? The variable should be very precisely defined.

   This study uses information of MODIS with some limitations in time and scale. In the "Material and methods" section we explain that product MOD09A1 has a spatial resolution of 500m x 500m and a time resolution of 8 days. In future researches we would like to prove with more spatial and time resolution using other products. Any case, we think we would obtain the same essential conclusion: Normal characterization is not the best in some intervals (mainly in spring and autumn).

4. I wonder how representative the presented study case is for the generalization of the results and conclusion. Different decisions made to develop this work should be justified: crop to evaluate, location, number of pixels, sensor… Why only some pixels and not the whole crop area?

   We have analyzed pasture in this study because this kind of crop uses NDVI characterization to define damage thresholds in the context of satellite index-base insurances. The selected location is an example of pasture area without trees dedicated to cattle breeding. You are right, this location is not very large and the spatial resolution of the MODIS product is low, so we were limited to use not much pixels. In future studies we want to select other more extensive pasture areas with more resolution for obtaining more relevant results.

5. The choice of the candidate functions (see Table 3) must also be justified.

   We have chosen these candidates because they are very common within asymmetrical distribution and we have shown in this study that observed NDVI distributions are essentially asymmetrical in many intervals (mainly in spring and autumn). We could have used other candidates, but we think the conclusion would be the same: normal assumption is not the best, and we recommend the use of quantiles.

6. Lines 441-444. This assertion is highly dependent on how representative the studied sample is of the case referred to, and to the general problem the paper wants to address.

   We have modified the paragraph eliminating the specific reference and talking about the Normal assumption methodology in general. Now you can read: "Therefore, the methodology using the NDVI Normal assumption applied to design an index-based insurance will not be feasible in many intervals of this study."

7. From the figures in the Annex it is not so clear that GEV has an overall better performance than the normal choice. In fact, in different examples both perform very similarly.

   You are right, GEV distributions fit better in spring and autumn intervals due to distributions are mainly asymmetric in these periods. We have included a new paragraph explaining this feature: Now you can read: "There is a relationship between seasons and the number of intervals that fit correctly. We found that GEV distributions explain better some intervals of spring and autumn since their observed distributions are very asymmetric. On the other hand, we did not find an important difference in winter, since its observed distributions are mainly symmetric".

8. I was expecting to find in the results more depth regarding the impact that a different distribution has on the parts (insurance companies and clients). At the end, even if statistically another function performs better than the normal assumption, the relevant issue is how much the benefit/loss is changed by this. A better performance might have only a minor impact on the final result in the insurance context. At least some estimation should be included.

   In this study we have focused in the statistical analysis and how the thresholds would be affected by the use of GEV assumption instead of the Normal one. We have also offered some estimation about the probability of being below the NDVI threshold in both assumptions. The probability obtained by GEV distributions is mostly lower than the Normal distributions in spring, autumn and winter, that is the working period of the insurance. In future works we will be able to focus in more economical aspects and to perform some simulation of the overall process.

9. Additionally, no discussion is done on other works in the results regarding the insurance context.

In this study we have shown up the differences in using Normal and GEV distributions in the insurance context. Differences in the probability of being below the NDVI threshold recommend us the use of quantiles instead of preset distributions.

10. Please, check that captions of tables and figures are self-explanatory (see for instance Table1; provide study period).

We have included the study period in the caption of table 1.

11. The weather variables statistics included in Table 1 just presents the local climate. Is it possible to combine this information with the NDVI monitoring to obtain better indexes or to validate the NDVI results?

Some studies have dealt with the combination of weather variables and NDVI to create a better index, however in this study we have focused in the statistical analysis of NDVI distributions without questioning whether NDVI is the best index or not.

12. Line 246. "… the completed station of meteorological networks". What do you mean by that?

We made a mistake with the translation. A completed station means a main or principal station with the majority of weather measure equipments.

13. Lines 306-311. What is the purpose of the last sentence?

We have not found the use of this HSL criterion in the context of NDVI remote sensing images. Therefore it can be considered a novelty method to eliminate wrong values in a NDVI series.

14. Line 367. Please, define this more precisely.

This entire paragraph has been rewritten to clarify all the definitions presented on it.

15. Line 371. I think the definition of VR can be more clearly expressed. Additionally, the use of tables for VR intervals results in too long and repeated content.

This entire paragraph has been rewritten to clarify all the definitions presented on it, and table 3 has been simplified.

16. Contents in lines 367-376 and 415-432 should be moved to the Methods sections.

We think some of these paragraphs could stay at the result section due to they are applications of the methodology presented in the Method section (MLM and Chi square test). We attend some of your suggestions and move the paragraph regarding to PDF candidates to the Method section.

17. Line 451. I would suggest not to use the future tense here.

We have modified from future tense to past tense.

18. Please, check the references' format meets the Journal's standards.

We have reviewed the format references.

[revised manuscript text omitted]

for each year. Therefore, 46 Random Variables (RV) were defined when taking into
account all the years of this study.
In Table 3, every RV (named as "Interval") can be seen together with the number of
available NDVI observations. Each RV collects the observations coming from the six
selected pixels. The start intervals of each season are: interval 45 for winter, interval 11
for spring, interval 23 for summer and interval 34 for autumn.

**Table 3.** Number of observations for every RV (named as Interval).

| RANDOM VARIABLE | # OBSERVATIONS | RANDOM VARIABLE | # OBSERVATIONS |
|---|---|---|---|
| Interval 1 | 85 | Interval 24 | 96 |
| Interval 2 | 84 | Interval 25 | 96 |
| Interval 3 | 96 | Interval 26 | 96 |
| Interval 4 | 96 | Interval 27 | 96 |
| Interval 5 | 95 | Interval 28 | 96 |
| Interval 6 | 90 | Interval 29 | 96 |
| Interval 7 | 86 | Interval 30 | 96 |
| Interval 8 | 83 | Interval 31 | 96 |
| Interval 9 | 96 | Interval 32 | 96 |
| Interval 10 | 96 | Interval 33 | 94 |
| Interval 11 | 74 | Interval 34 | 96 |
| Interval 12 | 88 | Interval 35 | 96 |
| Interval 13 | 88 | Interval 36 | 85 |
| Interval 14 | 88 | Interval 37 | 90 |
| Interval 15 | 96 | Interval 38 | 96 |
| Interval 16 | 92 | Interval 39 | 92 |
| Interval 17 | 88 | Interval 40 | 90 |
| Interval 18 | 96 | Interval 41 | 96 |
| Interval 19 | 95 | Interval 42 | 89 |
| Interval 20 | 96 | Interval 43 | 95 |
| Interval 21 | 95 | Interval 44 | 88 |
| Interval 22 | 96 | Interval 45 | 90 |
| Interval 23 | 96 | Interval 46 | 90 |

In Fig. 4, a plot with NDVI sample means of all RV with a start and end reference of
the astronomical seasons is shown. The typical evolution of the NDVI along a year can be
seen.

[Figure]

**Figure 4.** NDVI sample means of 46 random variables (RV) are shown as well as start and end
reference of every season. Study period from 2002 to 2017.

The observed evolution of NDVI through the different seasons is typical of the pasture
in this area. The summer presents the lowest mean values which begin to increase in
autumn achieving a maximum mean value of 0.60 or 0.65 during winter. In the middle of
the spring NDVI decrease again, approaching the lowest mean value of 0.28
approximately.

Taking into account these values, dense vegetation, in this study pasture, is found
from middle of October (interval 37) till the end of May (interval 19). It is in this period
where the precipitation concentrates (see Table 1). During the summer, the NDVI mean
values are lower than 0.3 corresponding with low precipitation and high temperatures.

Following the work of Escribano-Rodriguez et al. (2014), there is a relationship of
pasture damage and a NDVI value around 0.40. Even if the authors point out that this
value is highly variable depending on the location, we can see that summer season in this
case study is under this value (see Fig. 4). This can explain that "Insurances for Damaged
Pasture" usually do not apply in these dates due to the arid environment (BOE, 2013).

MLM has been applied to model these 46 RV. Parameters have been calculated for 4
PDF (see Table 2) which are the candidates to be the best fit. To check the goodness of the fit of PDF candidates, Chi square test (χ² test) has been used from 7 classes to 14 classes
meeting the requirement that each class has at least five observations. The level of
significance (α) was fixed to 5% for all the candidates.

Twelve intervals (from 23 to 34) corresponding to months of July, August and
September have been excluded of this analysis since these intervals fall into the dry
season in the study area, normally not cover by any SIBI. Therefore, calculations were
carried out over 34 intervals. Fig. 5 shows the percentage of intervals that fit for every PDF
candidate. The number of classes used in χ² test is represented at X-axis (from 7 to 14
classes).

[Figure]

**Figure 5.** Percentage of fitted intervals (Y axis) for each PDF candidate (Normal, Gamma, Beta and
GEV distributions) in function of the number of classes (X axis).

Fig. 5 indicates that GEV distributions explain more intervals (more than 40% for the
majority of the class analysis) than Normal, Gamma or Beta distributions. An important
difference between the Normal distribution and the rest of the PDF used in this work is its
symmetry and kurtosis. Many of the observed NDVI distributions present a clear
asymmetry and long tails in one or both sides that causes Normal distribution not to be
the optimal fit.

There is a relationship between seasons and the number of intervals that fit correctly.
We found that GEV distributions explain better some intervals of spring and autumn since
their observed distributions are very asymmetric. On the other hand, we did not find an
important difference in winter, since its observed distributions are mainly symmetric.

Therefore, the methodology using the NDVI Normal assumption applied to design an
index-based insurance will not be feasible in many intervals of this study.

[revised manuscript text omitted]

22, 237-247, 2016.

---

## Author Response (AR1)

**RESPONSES TO EDITOR:**

Thank you very much for all your suggestions and comments. Next, we respond all your questions in order:

1. All three reviewers question the selection of the GEV distribution and call for additional statistical metrics to justify this decision. The authors may wish to consider the use of relative quality estimators as additional statistical metrics to compare the different distribution models and corroborate the current results.

Answer: The quality estimator is the percentage of intervals that pass the $\chi^2$ test for every PDF candidate. For example: if we use $\chi^2$ test with 10 classes we obtain: GEV 41.2%, Beta 29.4%, Normal 23.5% and Gamma 17.6%. Therefore the best is GEV.

We have added the following clarifications.

Page 20, line 557 (in this document):

"The statistical metric used in this study to assess the fit of the observed NDVI values with respect to the PDF candidates (Normal, Gamma, Beta and GEV) was the Chi square test ($\chi^2$ test). The following steps were carried out:

1.  MLM was applied to model these 46 RV. Parameters were calculated for the four PDF candidates (see Table 2).
2.  To check the goodness of the fit of PDF candidates, Chi square test ($\chi^2$ test) was applied from 7 classes to 14 classes meeting the requirement that each class has at least five observations. The level of significance ($\alpha$) was fixed to 5% for all the candidates."

Page 21, line 599:

"Twelve intervals (from 23 to 34) corresponding to months of July, August and September have been excluded of this analysis since these intervals fall into the dry season in the study area, normally not cover by any SIBI. Therefore, calculations were carried out over 34 intervals.

To assess the general goodness of fit, the number of intervals where the $\chi^2$ test was accepted (or failed to reject) was calculated for every PDF candidate. Then, the percentage of accepted intervals, over the total 34 intervals, was also calculated (the quality estimator). Fig. 8 shows this percentage of intervals that fit for every PDF candidate. The number of classes used in $\chi^2$ test is represented at X-axis (from 7 to 14 classes)."

Our procedure has been to explore if a PDF could be used for a set of data respect to an interval. Sometimes all the PDF candidates could be used because all of them passed the $\chi^2$ test, other times only some of them. The best PDF candidate to be used along the year is the one with the highest percentage of intervals that passed the $\chi^2$ test (quality estimator). We are open to other quality estimator that the editor suggests. In any case, the aim of this study is not to prove that
GEV is the best possible fit, but to prove there are PDF candidates better than Normal.

2. Reviewer 3 calls for a literature review in the introduction (with additional references) on NDVI
distribution functions and limitations to the use of the normal distribution. Unfortunately these
issues have not been addressed in the interactive comments, but will definitely help the
formulation of discussion points.

We have added the following clarifications and references.

Page 7, line 213:

"Important NDVI-based indices of detecting drought are NDVI anomalies (NDVIA) and
Standardized Vegetation Index (SVI). NDVIA and SVI have been successfully used to monitor
drought conditions over different regions on the world (Nanzad et al., 2019; Li et al., 2014). NDVIA
is calculated as the difference between the NDVI value for a specific time period (e.g., week,
month) and the long-term mean value for that period. SVI was developed by Peters et al. (2002)
and obtains the probability from normal NDVI distributions over multiple years of data, on a time
period (Anyamba and Tucker, 2012; Bayarjargal et al., 2006). It is defined as:

$$SVI_i = \frac{NDVI_i - \overline{NDVI}}{\sigma_{NDVI}} = \frac{NDVIA_i}{\sigma_{NDVI}} \qquad (1)$$

where $\overline{NDVI}$ is the long-term mean NDVI in the period i, $\sigma_{NDVI}$ is the standard deviation of NDVI
in the period i, and $NDVI_i$ is the current NDVI value in the time period i. Using only the first and
second statistical moment, average and the square root of variance, assumption of normality is
implicit in this type of drought NDVI indicator."

If there are other references that we should include we will appreciate that you point out.

3. The manuscript lacks a separate discussion section: the authors should consider a split between
results and discussion. A separate dedicated section will help formulate strengths and weaknesses
of the study such as temporal, spatial and spectral scales, the representativeness for a wider area
and applicability to another environment. This section is necessary to place the research in a larger
context and relate the findings to other research.

Answer: We have split between results and discussion as it can be seen in the last version of the
manuscript.

4. On a more technical level, the following description may be added to address Reviewer 1's
comments on atmospheric correction. "Each MOD09A1 pixel contains the best possible L2G
observation during an 8-day period as selected on the basis of high observation coverage, low
view angle, the absence of clouds or cloud shadow, and aerosol loading." However, certain
observations were removed from further analysis, and therefore the question remains on what
basis these observations were removed.

Answer: We have added the description.

Page 11, line 364:

"Each MOD09A1 pixel contains the best possible L2G observation during an 8-day period as
selected on the basis of high observation coverage, low view angle, the absence of clouds or cloud
shadow, and aerosol loading."

5. I have serious concerns with respect to the (colour) filtering technique which seems to remove
all NDVI values below 0.2-0.25. This removal needs further explanation (or even exploration) in
view of the proposed extreme value distributions.

Answer: We have clarified the HSL filtering technique.

Page 16, line 490:

"MOD09A1 is a MODIS product that processes data to obtain the best observation in an 8-
days period. However, it is possible that the result of this selection still presents some problems
since the best of this selection is relative to the eight observations of the period. For example, if
the eight observations, at one pixel, appear with clouds, shadow clouds or snow, the best
selection still maintains this problem.
As an example of above, the NDVI series (10 years) of one pixel of the study area is shown in
Fig. 3. On the top graph of Fig. 3 it is noticed that there exit some extremely low NDVI values in
some dates. If these NDVI values are compared to neighbor values (8 days after or before) the high
variation presented in such short period is not believable. This issue tells us that MODIS sensor has
not obtained a proper observation in this 8 days period (interval).
HSL criterion helps us to eliminate these incorrect NDVI values, since the filter is interpreting
that these pixels still contains clouds or snow, i.e., pixels with low saturation (greyish colours)."

6. The mean NDVI profile presented in Figure 4 is very informative. However, an indication of
inter-quartile range would be even more informative, for instance in the form of a box plot. The
characterization of this seasonal variation and its explanation in agronomic terms seems crucial for the general understanding of the manuscript. The authors have the data to undertake this
analysis.

Answer: We have added boxplots in Figure 4.

Page 19:

[revised manuscript text omitted]

22, 237-247, 2016.

---

## Editor Decision (ED1)

Editor's report on "Statistical Analysis for Satellite Index-Based Insurance to define Damaged Pasture Thresholds" - Juan José Martín-Sotoca et al.

The authors present an analysis of satellite index based distributions across the growing season as a basis for defining pasture damage. The topic is very relevant not only to the journal but also within the broader context of agricultural insurances in a changing climate. Overall the research is well presented and structured.

The reports of three reviewers document major shortcomings in the current manuscript. The interactive comments in reply to the reviews do not always reflect the call for revision sufficiently; nor do the suggested changes in the manuscript. Some comments deserve better attention and subsequent changes should be reflected in the manuscript. In my opinion, the authors have enough data and experience to deal with the comments in a better way so that the resulting manuscript will attract a larger audience.

The following major points will require the authors' attention prior to publication:

1. All three reviewers question the selection of the GEV distribution and call for additional statistical metrics to justify this decision. The authors may wish to consider the use of relative quality estimators as additional statistical metrics to compare the different distribution models and corroborate the current results.

2. Reviewer 3 calls for a literature review in the introduction (with additional references) on NDVI distribution functions and limitations to the use of the normal distribution. Unfortunately these issues have not been addressed in the interactive comments, but will definitely help the formulation of discussion points.

3. The manuscript lacks a separate discussion section: the authors should consider a split between results and discussion. A separate dedicated section will help formulate strengths and weaknesses of the study such as temporal, spatial and spectral scales, the representativeness for a wider area and applicability to another environment. This section is necessary to place the research in a larger context and relate the findings to other research.

4. On a more technical level, the following description may be added to address Reviewer 1's comments on atmospheric correction. "Each MOD09A1 pixel contains the best possible L2G observation during an 8-day period as selected on the basis of high observation coverage, low view angle, the absence of clouds or cloud shadow, and aerosol loading." However, certain observations were removed from further analysis, and therefore the question remains on what basis these observations were removed.

5. I have serious concerns with respect to the (colour) filtering technique which seems to remove all NDVI values below 0.2-0.25. This removal needs further explanation (or even exploration) in view of the proposed extreme value distributions.

6. The mean NDVI profile presented in Figure 4 is very informative. However, an indication of inter-quartile range would be even more informative, for instance in the form of a box plot. The characterization of this seasonal variation and its explanation in agronomic terms seems crucial for the general understanding of the manuscript. The authors have the data to undertake this analysis.

The above points need careful revision and therefore I propose a major revision.

---

## Author Response (AR2)

**RESPONSES TO EDITOR:**

Thank you very much for all your technical corrections and comments. Next, we add the marked manuscript with all your corrections:

[revised manuscript text omitted]

22, 237-247, 2016.